# MULTI-GRAPH META-TRANSFORMER (MGMT)

## ABSTRACT

Multi-graph learning is crucial for extracting meaningful signals from collections of heterogeneous graphs. However, effectively integrating information across graphs with differing topologies, scales, and semantics, often in the absence of shared node identities, remains a significant challenge. We present the Multi-Graph Meta-Transformer (MGMT), a unified, scalable, and interpretable framework for cross-graph learning. MGMT first applies Graph Transformer encoders to each graph, mapping structure and attributes into a shared latent space. It then selects task-relevant supernodes via attention and builds a meta-graph that connects functionally aligned supernodes across graphs using similarity in the latent space. Additional Graph Transformer layers on this meta-graph enable joint reasoning over intra- and inter-graph structure. The meta-graph provides built-in interpretability: supernodes and superedges highlight influential substructures and cross-graph alignments. Evaluating MGMT on both synthetic datasets and real-world neuroscience applications, we show that MGMT consistently outperforms existing state-of-the-art models in graph-level prediction tasks while offering interpretable representations that facilitate scientific discoveries. Our work establishes MGMT as a unified framework for structured multi-graph learning, advancing representation techniques in domains where graph-based data plays a central role.

## 1 INTRODUCTION

Graphs are fundamental data structures in many domains including neuroscience (Shahbaba et al., 2022; Zhou et al., 2024), social networks (Fan et al., 2019; Zhang et al., 2022) and molecular biology (Wieder et al., 2020; Xu et al., 2022; Li et al., 2022). While powerful models like Graph Neural Networks (GNNs) (Scarselli et al., 2008; Kipf & Welling, 2016; Yu et al., 2023) and the more recent Graph Transformers (GTs) (Wu et al., 2021; Kreuzer et al., 2021; Rampášek et al., 2022; Kim et al., 2022) excel at learning from single graphs, many real-world problems require integrating information across multiple heterogeneous graphs, where the heterogeneity may stem from differences in modalities, views, or population characteristics. For instance, neuroscience experiments studying brain dynamics often generate graphs from multiple subjects, each with distinct connectivities and node sets (Shahbaba et al., 2022). Enhancing prediction performance or extracting common neural patterns in such settings requires a framework capable of effectively integrating information across structurally heterogeneous graphs. Nonetheless, the question of how to optimally adapt powerful architectures such as the GT to the multi-graph integration problem remains an active area of research, particularly with heterogeneous structures, unaligned node sets, and a need for fine-grained cross-graph reasoning, conditions that arise in many scientific domains.

Recent work on multi-graph learning partially addresses this challenge by either operating on a single unified heterogeneous graph with aligned nodes (He et al., 2025; Zhang et al., 2019; Zheng et al., 2022) or by learning graph-level embeddings from multiple modality-specific graphs and combining them via shared contexts, adaptive weights, or correlation-based objectives (Hayat et al., 2022; Xu et al., 2024; Xing et al., 2024; D'Souza et al., 2023; Nakhli et al., 2023; Fu et al.). However, these methods model cross-graph interactions only through pooled graph embeddings or shared tokens, without explicit fine-grained message passing between structurally similar subgraphs across unaligned graphs, which limits interpretability and makes it challenging to identify which substructures across graphs interact and contribute to the model's predictions.

To address these limitations, we propose the *Multi-Graph Meta-Transformer* (MGMT), a flexible framework for integrating information across collections of heterogeneous graphs. Under the umbrella

term "multi-graph," our unified approach accommodates common scenarios, including: **multimodal** (graphs from different measurement channels), **multi-view** (different structural views of the same data), and **multi-subject** (graphs from heterogeneous subjects in the same experiment). MGMT first processes each input graph using a graph-transformer encoder and aggregates intermediate layer outputs through a depth-aware mixing scheme. This yields node embeddings that adaptively integrate information across multiple receptive-field sizes. The framework is backbone-agnostic and can be implemented with either localized or global GT layers, depending on the task and computational budget. It then selects a small set of informative *supernodes* from each graph using attention scores and constructs an explicit meta-graph over these task-relevant substructures, preserving within-graph connectivity while selectively introducing cross-graph (inter-graph) edges between functionally aligned regions. Additional Transformer layers on the resulting meta-graph perform fine-grained cross-graph message passing and generate the final prediction.

This design directly addresses the key limitations of existing multi-graph approaches. Instead of representing each graph solely through global graph-level or context-level embeddings at fusion time, MGMT preserves structure at the node and subgraph levels and performs fusion by attending over an explicit meta-graph of supernodes, encoding both intra-graph and inter-graph structure. More specifically, supernodes summarize local patterns within each graph, while superedges align substructures across graphs and capture their interactions. The resulting meta-graph also provides insights into task-relevant subgraphs and their relationships, which can help with interpreting the results. Taken together, MGMT utilizes depth-aware, structure-preserving GT encoders within each graph, identifies supernodes, and uses them to build an explicit meta-graph that supports cross-graph message passing and provides a principled and scalable approach to aggregating information across a collection of heterogeneous graphs with unaligned node sets.

Our main contributions are as follows: (1) For heterogeneous graphs (multimodal, multi-view, or multi-subject), we formalize a data-fusion framework, MGMT, which provides a backbone-agnostic, depth-aware, and structure-preserving architecture with interpretable outputs through an explicitly constructed meta-graph over attention-selected supernodes; (2) we provide theoretical results demonstrating that MGMT's depth-aware aggregation can recover general $L$-hop neighborhood mixing and characterize conditions under which the induced meta-graph function class offers strictly improved approximation capacity relative to late-fusion strategies that operate only on pooled graph embeddings; (3) using synthetic benchmarks and real-world applications, we show that MGMT outperforms state-of-the-art multimodal and graph-based methods, including recent multi-graph and transformer architectures; and (4) we demonstrate that MGMT can detect meaningful neurobiological patterns, thereby offering insights for scientific investigations, particularly for understanding neural mechanisms underlying memory and factors contributing to Alzheimer's disease.

## 2 RELATED WORK

**Graph Representation Learning.** GNN is the cornerstone of modern graph machine learning, which learns node embeddings via local message passing (Scarselli et al., 2008; Kipf & Welling, 2016; Velickovic et al., 2017). Attention-based models such as GAT (Velickovic et al., 2017) learn neighbor-specific attention weights instead of using fixed aggregation rules, allowing them to prioritize more informative neighbors. Nevertheless, because they still aggregate information only from local neighborhoods, their ability to distinguish graph structures remains fundamentally limited by the expressive power of the 1-Weisfeiler–Lehman (1-WL) test. More recent GT architectures with structured self-attention have outperformed message-passing GNNs on a variety of benchmarks (Dwivedi & Bresson, 2020; Vaswani, 2017). Global-attention models such as EGT (Hussain et al., 2022) replace or complement convolution with fully-connected self-attention augmented with structural encodings, thereby extending expressivity. Sparse and structure-aware GTs, including Exphormer (Shirzad et al., 2023) and GRIT (Ma et al., 2023), introduce scalable attention patterns and graph inductive biases that retain or approximate the expressivity of dense Transformers while reducing computational cost. Hierarchical and distance-structured GT variants, such as HDSE (Luo et al., 2024), further refine how multi-scale structural information is injected into attention. All these existing methods, however, lack a principled and interpretable mechanism for fusing multiple heterogeneous graphs, which are common in many scientific applications. MGMT addresses this gap by providing a flexible and general framework; in Appendix A12, we additionally implement MGMT with several of these GT backbones and compare their performance in both single-graph and multi-graph settings.

**Heterogeneous Unified-Graph Representation Learning.** A related line of research is multimodal or heterogeneous graph learning, which may appear similar to our multi-graph fusion task but is fundamentally different, as it operates on a single unified graph that integrates diverse data types. For example, frameworks like UniGraph2 (He et al., 2025) and HetGNN (Zhang et al., 2019) assume a single graph where nodes possess multiple feature types from different modalities, such as text or images. The common approach is to collapse multiple data sources into one large graph. Other works like MMGL (Zheng et al., 2022) construct a single population-level graph where nodes represent subjects, and features from all modalities are concatenated before graph construction. While effective for their intended purpose, they are not applicable to the more challenging problem of fusing a collection of graphs with distinct, unaligned node sets, which is the focus of our work.

**General-Purpose Multimodal Fusion.** Models such as MultiMoDN (Swamy et al., 2023), Flex-Care (Xu et al., 2024), MedFuse (Hayat et al., 2022), and Meta-Transformer (MT) (Ma et al., 2022) aim to integrate multiple modalities, including graphs or images. In principle, these frameworks could be applied to multi-graph fusion by treating each graph as a separate modality. Typically, they use modality-specific encoders to transform each input into a single latent vector. For a graph, this amounts to collapsing its entire topological structure into one embedding. These vectors are then fused, for example via concatenation, for downstream tasks. While highly effective in many multimodal settings, such vector-level fusion largely ignores graph topology and subgraph-level relationships across multiple graphs, which limits interpretability.

**Multi-graph learning.** Another class of models focuses on settings where each entity is associated with multiple graphs. Recent multi-graph models such as AMIGO (Nakhli et al., 2023), EMO-GCN (Xing et al., 2024), and MaxCorrMGNN (D'Souza et al., 2023) take multiple graph-structured inputs but ultimately fuse information only at the level of pooled graph embeddings or shared context tokens, so cross-graph interaction is mediated through global representations instead of local structural alignment. MGLAM (Fu et al.), on the other hand, treats each entity as a bag of graphs and learns permutation-invariant bag-level predictors via kernel-based graph representations and multi-graph pooling, providing a principled baseline for the multi-graph-to-label setting considered in this work. However, all of these models still perform fusion at the graph/bag level, as opposed to using explicit node-level cross-graph message passing. This limits their ability to explain how specific nodes or substructures interact across graphs and how particular cross-graph patterns influence predictions.

Beyond the above per-entity multi-graph models, another class of multi-graph and multi-view GTs, such as MGT and MVGT/MVGTrans (Cui et al., 2024; Zhou et al., 2025), assumes multiple edge views over a shared node set but is not directly applicable to heterogeneous graph collections with disjoint node sets across modalities or subjects. An alternative approach involves learning consensus graphs or dataset-level representations rather than per-entity meta-graphs. This includes AMGL (Nie et al., 2016), GraphFM and GraphAlign (Lachi et al., 2024; Hou et al., 2024), which aggregate information across graphs at the population level, while graph-of-graphs models such as SamGoG (Wang et al., 2025) propagate information solely at the graph-instance level. In contrast to both graph-level fusion and population-level aggregation, MGMT constructs a meta-graph whose nodes are attention-selected supernodes drawn from all graphs of an entity, preserving intra-graph connectivity while enabling fine-grained, node-level cross-graph message passing. This approach leads to more interpretable results, as it reveals how different graphs interact to drive final prediction.

## 3 METHODOLOGY

In this section, we present MGMT, detailing its prediction pipeline based on GT encoders and meta-graph construction, followed by describing how to interpret MGMT by identifying significant nodes and edges in Section 3.2. An overview of the entire framework is provided in Fig. 1.

### 3.1 MULTI-GRAPH META-TRANSFORMER (MGMT)

MGMT fuses multi-graph data using several steps, as described below.

#### 3.1.1 GRAPH-SPECIFIC TRANSFORMER ENCODERS

For each instance, we observe a collection of $n$ graphs. For $i = 1, \ldots, n$, we denote the graph as $\mathcal{G}_i = (\mathcal{V}_i, \mathcal{E}_i)$ with node set $\mathcal{V}_i$ of size $N_i = |\mathcal{V}_i|$, and edge set $\mathcal{E}_i$. Each graph $\mathcal{G}_i$ is characterized by

a node feature matrix $\boldsymbol{X}_i \in \mathbb{R}^{N_i \times d}$ and an adjacency matrix $\boldsymbol{A}_i \in \{0,1\}^{N_i \times N_i}$. Graphs per each instance may differ in size and structure (for presentation purposes only, we assume feature size is $d$ across all graphs), yet the collection $\{\mathcal{G}_1, \ldots, \mathcal{G}_n\}$ share a common label $Y \in \mathcal{Y}$. The task is graph-level classification of the shared label $Y$ using evidence aggregated across graphs. Throughout this paper, we use bold uppercase letters (e.g., $\boldsymbol{X}$) for matrices and bold lowercase letter (e.g., $\boldsymbol{x}$) for vectors, and $[n]$ denoting the set $\{1, \ldots, n\}$.

We formalize the core graph-specific Transformer mechanics used in MGMT, building upon the localized graph-aware attention principles detailed in Appendix A1. For each $i \in [n]$, the graph $\mathcal{G}_i$ with node features $\boldsymbol{X}_i \in \mathbb{R}^{N_i \times d}$ undergoes $L$ GT layers with multi-head self-attention. Starting with $\boldsymbol{H}_i^{(0)} = \boldsymbol{X}_i$ as initial features, we define the extended neighborhood $\bar{\mathcal{N}}(u) = \mathcal{N}(u) \cup \{u\}$ to ensure nodes attend to themselves during message passing.

For layer $\ell \in [L]$, attention head $m \in [M]$, and edge $(u,v) \in \mathcal{E}_i \cup \{(u,u)\}$, we compute:

$$
\begin{aligned}
\boldsymbol{Q}_{i,u}^{(\ell,m)} &= \boldsymbol{W}_{Q,i}^{(\ell,m)} \boldsymbol{H}_{i,u}^{(\ell-1)} + \boldsymbol{b}_{Q,i}^{(\ell,m)}, & \alpha_{i,uv}^{(\ell,m)} &= \frac{\exp\left(\boldsymbol{Q}_{i,u}^{(\ell,m)\top} \boldsymbol{K}_{i,v}^{(\ell,m)} / \sqrt{d'}\right)}{\sum_{v' \in \bar{\mathcal{N}}(u)} \exp\left(\boldsymbol{Q}_{i,u}^{(\ell,m)\top} \boldsymbol{K}_{i,v'}^{(\ell,m)} / \sqrt{d'}\right)}, \\
\boldsymbol{K}_{i,v}^{(\ell,m)} &= \boldsymbol{W}_{K,i}^{(\ell,m)} \boldsymbol{H}_{i,v}^{(\ell-1)} + \boldsymbol{b}_{K,i}^{(\ell,m)}, & & \\
\boldsymbol{V}_{i,v}^{(\ell,m)} &= \boldsymbol{W}_{V,i}^{(\ell,m)} \boldsymbol{H}_{i,v}^{(\ell-1)} + \boldsymbol{b}_{V,i}^{(\ell,m)}, & \boldsymbol{Z}_{i,u}^{(\ell,m)} &= \sum_{v \in \bar{\mathcal{N}}(u)} \alpha_{i,uv}^{(\ell,m)} \boldsymbol{V}_{i,v}^{(\ell,m)},
\end{aligned}
\tag{1}
$$

where $\boldsymbol{H}_{i,u}^{(\ell-1)} \in \mathbb{R}^d$ is the feature of node $u$ at layer $\ell-1$, $d' = d/M$ denotes the per-head dimension. Projection matrices $\boldsymbol{W}_{Q,i}^{(\ell,m)}, \boldsymbol{W}_{K,i}^{(\ell,m)}, \boldsymbol{W}_{V,i}^{(\ell,m)} \in \mathbb{R}^{d' \times d}$ and biases $\boldsymbol{b}_{Q,i}^{(\ell,m)}, \boldsymbol{b}_{K,i}^{(\ell,m)}, \boldsymbol{b}_{V,i}^{(\ell,m)} \in \mathbb{R}^{d'}$ are learnable parameters. The query vector $\boldsymbol{Q}_{i,u}^{(\ell,m)}$ represents information node $u$ seeks from neighbors, key vector $\boldsymbol{K}_{i,v}^{(\ell,m)}$ encodes neighbor $v$'s relevance, and value vector $\boldsymbol{V}_{i,v}^{(\ell,m)}$ contains content to be aggregated. Attention score $\alpha_{i,uv}^{(\ell,m)}$ determines how much node $u$ attends to node $v$.

The outputs of all heads are concatenated ($\|$ denotes the concatenation) and transformed via:

$$
\boldsymbol{Z}_{i,u}^{(\ell)} = \big\|_{m \in [M]} \left[ \boldsymbol{Z}_{i,u}^{(\ell,1)}, \ldots, \boldsymbol{Z}_{i,u}^{(\ell,M)} \right] \boldsymbol{W}_{O,i}^{(\ell)} + \boldsymbol{b}_{O,i}^{(\ell)},
$$

where $\boldsymbol{W}_{O,i}^{(\ell)} \in \mathbb{R}^{d \times d}$, $\boldsymbol{b}_{O,i}^{(\ell)} \in \mathbb{R}^d$. Stacking these vectors across all nodes yields $\boldsymbol{Z}_i^{(\ell)} \in \mathbb{R}^{N_i \times d}$.

$\boldsymbol{Z}_i^{(\ell)}$ is processed by an FFN with activation, then combined via residual and LayerNorm to yield:

$$
\boldsymbol{H}_i^{(\ell)} = \text{LayerNorm}(\boldsymbol{Z}_i^{(\ell)} + \sigma(\text{FFN}(\boldsymbol{Z}_i^{(\ell)})))
\tag{2}
$$

After $L$ layers, we obtain final output and attentions by dynamically aggregating across all depths:

$$
\begin{aligned}
\boldsymbol{H}_i &= \sum_{\ell \in [L]} \Gamma_i^{(\ell)} \boldsymbol{H}_i^{(\ell)} \in \mathbb{R}^{N_i \times d}, \\
\boldsymbol{\alpha}_i &= \left\{ \alpha_{i,uv} = \sum_{\ell \in [L]} \Gamma_i^{(\ell)} \left( \frac{1}{M} \sum_{m \in [M]} \alpha_{i,uv}^{(l,m)} \right) \right\}_{(u,v) \in \mathcal{E}_i \cup \{(u,u)\}},
\end{aligned}
\tag{3}
$$

$\{\Gamma_i^{(\ell)}\}_{\ell=1}^n$ are confidence scores measuring quality of each Transformer layer (Section A2 for details).

### 3.1.2 SUPERNODE EXTRACTION

To identify the most informative nodes in each graph $i$, we extract *supernodes* based on the learned attention scores $\boldsymbol{\alpha}_i$ in equation 3. Given a predefined threshold $\tau$, we form the set of supernodes as

$$
\mathcal{S}_i = \left\{ u \in \mathcal{V}_i \mid \sum_{(u,v) \in \mathcal{E}_i} \alpha_{i,uv} \geq \tau \right\}.
\tag{4}
$$

Intuitively, $\sum_{(u,v) \in \mathcal{E}_i} \alpha_{i,uv}$ quantifies the total attention distributed by node $u$ to its neighbors.

We then induce a subgraph over these nodes:

$$
\mathcal{G}_i' = (\mathcal{S}_i, \mathcal{E}_i'), \mathcal{E}_i' = \{(u,v) \in \mathcal{E}_i \mid u, v \in \mathcal{S}_i\}
\tag{5}
$$

We conduct a sensitivity study in Section A10 to examine how choices of $\tau$ influence performance. Our analysis reveals that $\tau$ controls a trade-off: a higher $\tau$ creates a sparser meta-graph, which risks information loss, while a lower $\tau$ retains more nodes, risking overfitting. By guiding the selection of $\tau$ via cross-validation, we identified a robust range of values that yields stable performance.

### 3.1.3 META-GRAPH CONSTRUCTION

To model both intra- and cross-graph interactions, we construct an instance-level meta-graph $\mathcal{G}_M = (\mathcal{S}_M, \mathcal{E}_M)$, where $\mathcal{S}_M = \bigcup_{i=1}^{n} \mathcal{S}_i$ contains all graph-level supernodes. Each $u \in \mathcal{S}_i$ is associated with a latent embedding $\boldsymbol{H}_{i,u} \in \mathbb{R}^d$ as defined in equation 3. The edge set $\mathcal{E}_M$ of the meta-graph includes two components. First, we retain all intra-graph edges from the pruned graphs $\mathcal{G}'_i = (\mathcal{S}_i, \mathcal{E}'_i)$, preserving graph-specific relationships. Second, we introduce inter-graph edges between cross-graph supernodes using their feature similarity. For any node pair $(u, v)$ with $u \in \mathcal{S}_i$, $v \in \mathcal{S}_j$, and $i \neq j$, we compute the cosine similarity:

$$e_{uv} = \frac{\boldsymbol{H}_u^\top \boldsymbol{H}_v}{\|\boldsymbol{H}_u\|\|\boldsymbol{H}_v\|} \tag{6}$$

If the similarity score $e_{uv}$ exceeds a predefined threshold $\gamma$, the edge $(u, v)$ is added to $\mathcal{E}_M$.

Figure 1: Architecture of Multi-Graph Meta-Transformer (MGMT). Depth-Aware GT layers process individual graphs, extracting supernodes to form a meta-graph. Additional GT layers model both intra- and inter-graph interactions.

The resulting adjacency matrix $\boldsymbol{A}_M \in \mathbb{R}^{|\mathcal{S}_M| \times |\mathcal{S}_M|}$ encodes both intra- and inter-graph relationships among supernodes. We connect supernodes across graphs only when their latent embeddings are similar, mirroring observations that learning or selecting edges to reduce Dirichlet energy improves downstream accuracy Chen et al. (2020); see Section A5 for the formal smoothness justification. Appendix A10 shows accuracy is non-monotone in $\gamma$, reflecting the trade-off between dense connectivity and sparsity. Appendix A11 shows comparable performance with cosine, Pearson, Euclidean, and dot-product similarities, indicating robustness to the similarity choice. In AppendixA13, we replace the validation-tuned thresholds $\tau$ and $\gamma$ with dynamic, distribution-based quantile thresholds and show that this data-driven variant achieves comparable accuracy with the best validation-tuned threshold configuration on all datasets, indicating that MGMT is robust to threshold selection and does not hinge on delicate manual tuning.

### 3.1.4 FEATURE LEARNING AND PREDICTION

After constructing $\mathcal{G}_M$, we apply additional GT layers to the stacked supernode embeddings $\boldsymbol{H}_M^{(0)} \in \mathbb{R}^{|\mathcal{S}_M| \times d}$. Multi-head self-attention and feedforward updates are applied to capture global contextual dependencies, resulting in updated supernode embeddings $\boldsymbol{H}_M \in \mathbb{R}^{|\mathcal{S}_M| \times d}$. For classification, we apply permutation-invariant pooling followed by a fully connected network: $\hat{y} = f(\text{Pool}(\boldsymbol{H}_M))$. $\text{Pool}(\cdot)$ is pooling/aggregation operator, and $f(\cdot)$ maps pooled vector to class probabilities $\hat{y} \in \mathbb{R}^{|\mathcal{Y}|}$.

### 3.2 INTERPRETATION OF MGMT

The identified meta-graph is analyzed via **Node-level analysis**, highlighting influential nodes and their contributions, and **Edge-level analysis**, uncovering critical relationships among these nodes. This framework enhances interpretability, as illustrated in neuroscience application results in Section 5.3.

## 4 THEORETICAL PROPERTIES

In this section, we establish MGMT's theoretical foundations through: (1) *intra-graph analysis*, demonstrating superior feature representation within individual graphs; and (2) *inter-graph analysis*, showing enhanced predictive power through meta-graph construction. Complete proofs are provided in Section A3, with additional theoretical results in Section A4.

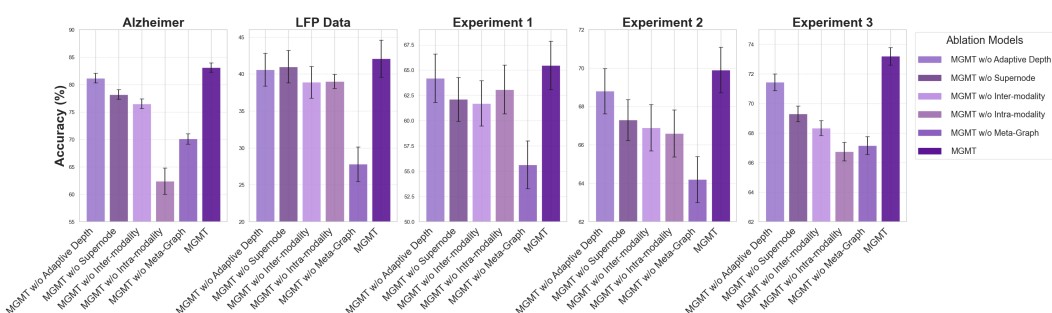

Figure 2: Ablations on five datasets. Dropping adaptive depth, supernode selection, or inter-modality edges lowers accuracy, confirming each component's importance for cross-graph learning.

## 4.1 INTRA-GRAPH ANALYSIS

We analyze the depth-aware mixing strategy in equation 3 which enables MGMT to aggregate information across different depths of message passing. First, we establish some formal definitions.

Let $\mathcal{M}(\boldsymbol{A}) \in \mathbb{R}^{N \times N}$ be a message-passing operator (e.g., augmented adjacency matrix, $\mathcal{M}(\boldsymbol{A}) = \boldsymbol{A} + \boldsymbol{I}$). Given $\mathcal{M}(\boldsymbol{A})$ and an activation function $\sigma$, denote the 1-hop feature aggregation as

$$\mathcal{U}(\boldsymbol{X}; \mathcal{M}(\boldsymbol{A}), \sigma) := \sigma(\mathcal{M}(\boldsymbol{A})\boldsymbol{X}),$$

and the $\ell$-hop aggregation is the $\ell$-fold composition of $\mathcal{U}$, namely,

$$\mathcal{U}^{\ell}(\boldsymbol{X}; \mathcal{M}(\boldsymbol{A}), \sigma) := \underbrace{\sigma(\mathcal{M}(\boldsymbol{A}) \cdots \sigma(\mathcal{M}(\boldsymbol{A})}_{\ell \text{ times}} \boldsymbol{X})).$$

Building on these, we introduce $L$-**hop mixing**, characterizing a model's ability to represent multi-depth information. While originally studied for Graph Convolutional Networks with graph Laplacians (Abu-El-Haija et al., 2019), we extend this concept to general message passing operators.

**Definition 4.1** ($L$-hop mixing with general message passing). *Given $\mathcal{M}(\cdot)$, a model is capable of representing $L$-hop mixing if for any $\eta_1, \ldots, \eta_L \in \mathbb{R}$, there exists a setting of its parameter and an injective (one-to-one) mapping $f(\cdot)$, such that the output of the model is equivalent as*

$$f\left(\sum_{\ell=1}^{L} \eta_{\ell} \cdot \mathcal{U}^{\ell}(\boldsymbol{X}; \mathcal{M}(\boldsymbol{A}), \sigma)\right), \tag{7}$$

*for any adjacency matrix $\boldsymbol{A}$, activation function $\sigma$, and node features $\boldsymbol{X}$.*

**Remark 4.2.** *If $\mathcal{M}(\boldsymbol{A}) = \boldsymbol{D}^{-\frac{1}{2}}(\boldsymbol{A} + \boldsymbol{I})\boldsymbol{D}^{-\frac{1}{2}}$, where $\boldsymbol{D}$ is the diagonal degree matrix with $D_{ii} = \sum_{j=1}^{N} A_{ij} + 1$, Definition 4.1 recovers the $L$-hop mixing with Graph Laplacian in the GCN literature (Abu-El-Haija et al., 2019; Zhou et al., 2024).*

First theorem demonstrates that MGMT's depth-aware GTs represent $L$-hop mixing for each graph.

**Theorem 4.3.** *With message passing operator $\mathcal{M}(\boldsymbol{A}) = \text{softmax}(\boldsymbol{A} + \boldsymbol{I})$, where softmax is applied row-wise. MGMT's depth-aware GTs in equation 1–equation 3 can represent $L$-hop mixing.*

The proof appears in Section A3. Notably, we also demonstrate in Section A4.1 that vanilla Graph Transformers **cannot** learn $L$-hop neighborhood mixing. We further clarify the relationship between $L$-hop mixing and Weisfeiler-Leman expressivity in Section A4.3, showing that these characterize distinct but complementary aspects of model power.

## 4.2 INTER-GRAPH ANALYSIS

This section analyzes how MGMT's meta-graph construction boosts prediction power compared to late fusion approaches (Zhang et al., 2023).

Recall from Section 3.1.3, the meta-graph $\mathcal{G}_M = (\mathcal{S}_M, \mathcal{E}_M)$ combines supernodes $\mathcal{S}_M = \bigcup_{i=1}^{n} \mathcal{S}_i$. Its initial embedding $\boldsymbol{H}_M^{(0)} \in \mathbb{R}^{|\mathcal{S}_M| \times d}$ stacks supernode embeddings where $\forall u \in \mathcal{S}_i, \boldsymbol{H}_{M,u}^{(0)} = \boldsymbol{H}_{i,u}$.

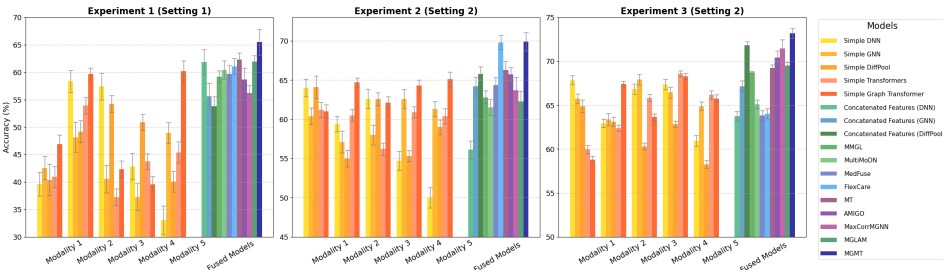

Figure 3: Average test accuracy and standard error bars (computed over 50 repetitions) on three synthetic datasets. In all experiments, each sample consists of five synthetic graphs, which we refer to as Modalities 1–5. Experiment 1 (Setting 1) uses 100 samples, with 5 nodes per graph, all of which are informative. Experiments 2 and 3 (Setting 2) both involve structured noise: Experiment 2 uses 100 samples and Experiment 3 uses 2,000 samples; in both, each graph has 50 nodes, of which 40 are informative. Across all configurations, the proposed MGMT model achieves the best performance.

MGMT applies additional $L_{\text{GT}}$ Graph Transformer layers followed by a global pooling to obtain the final graph-level embedding. Lastly, we apply $L_{\text{MLP}}$ MLP layers for class probabilities. Assume without loss of generality that $L_{\text{GT}} = 1$ and $L_{\text{MLP}} = 2$, the function class of MGMT given $\boldsymbol{H}_M^{(0)}$ is

$$\mathcal{F}_M = \left\{ f : \mathbb{R}^{|\mathcal{S}_M| \times d} \mapsto \mathbb{R}^{|\mathcal{Y}|} \ \Big| \ f = \boldsymbol{W}_{\text{MLP}}^{(2)} \sigma\left(\boldsymbol{W}_{\text{MLP}}^{(1)} \text{Pool}(\text{GT}(\boldsymbol{H}_M^{(0)}))\right) \right\}, \tag{8}$$

where $\text{GT}(\cdot) : \mathbb{R}^{|\mathcal{S}_M| \times d} \mapsto \mathbb{R}^{|\mathcal{S}_M| \times d}$ is the Graph Transformer, $\text{Pool}(\cdot) : \mathbb{R}^{|\mathcal{S}_M| \times d} \mapsto \mathbb{R}^{h'}$ is a graph pooling, and $\boldsymbol{W}_{\text{MLP}}^{(1)} \in \mathbb{R}^{h' \times h''}$, $\boldsymbol{W}_{\text{MLP}}^{(2)} \in \mathbb{R}^{|\mathcal{Y}| \times h''}$ are MLP weight matrices, with $h', h'' \in \mathbb{N}^+$. All subsequent analysis could be easily extended to any number of $L_{\text{MLP}}$ and $L_{\text{GT}}$.

The late fusion strategy that employs weighted averaging of class probabilities from graph-specific models can be represented as

$$\mathcal{F}_{\text{late}} = \left\{ f : \mathbb{R}^{|\mathcal{S}_M| \times d} \mapsto \mathbb{R}^{|\mathcal{Y}|} \ \Big| \ f = \sum_{i=1}^{n} w_i \cdot \boldsymbol{W}_{\text{MLP},i}^{(2)} \sigma\left(\boldsymbol{W}_{\text{MLP},i}^{(1)} \text{Pool}_{\mathcal{S}_i}(\boldsymbol{H}_M^{(0)})\right) \right\},$$

where $\{\boldsymbol{W}_{\text{MLP},i}^{(\ell)}\}_{l \in [2], i \in [n]}$ is the set of graph-specific MLP parameter, and the set of late fusion weights is $\{w_i \in \mathbb{R}\}_{i \in [n]}$ such that $\sum_{i=1}^{n} w_i = 1$. Given the joint distribution of a feature-label pair $(\boldsymbol{X}, Y) \sim \mathcal{P}$ and a loss function $\mathcal{L}$, denote the generalization error of a function $f$ as

$$R(f; \mathcal{P}, \mathcal{L}) \coloneqq \mathbb{E}_{(\boldsymbol{X}, Y) \sim \mathcal{P}}[\mathcal{L}(f(\boldsymbol{X}), Y)]$$

Following Shalev-Shwartz & Ben-David (2014), we define the **approximation error** of a function class $\mathcal{F}$ as the minimum generalization error achievable by a function in $\mathcal{F}$, namely,

$$\epsilon(\mathcal{F}; \mathcal{P}, \mathcal{L}) \coloneqq \inf_{f \in \mathcal{F}} R(f; \mathcal{P}, \mathcal{L}). \tag{9}$$

Assume latent representations of the meta graph follow $(\boldsymbol{H}_M^{(0)}, Y) \sim \mathcal{P}_M$. The next theorem shows MGMT is a more powerful graph fusion framework compared to late fusion in the sense that it achieves a smaller approximation error.

**Theorem 4.4.** *Denote approximation error of MGMT on the meta-graph as $\epsilon(\mathcal{F}_M; \mathcal{P}_M, \mathcal{L})$, and the approximation error of late fusion of graph-specific classifiers $\epsilon(\mathcal{F}_{\text{late}}; \mathcal{P}_M, \mathcal{L})$, then*

$$\epsilon(\mathcal{F}_M; \mathcal{P}_M, \mathcal{L}) \leq \epsilon(\mathcal{F}_{\text{late}}; \mathcal{P}_M, \mathcal{L}).$$ See Section A4 for the proof.

## 5 NUMERICAL EXPERIMENTS

We evaluate the effectiveness of MGMT on three synthetic datasets in Section 5.2 and two real-world neuroscience applications in Section 5.3 (memory experiment and Alzheimer's disease detection). The synthetic and Alzheimer's datasets are multi-modal, whereas the memory experiment is multi-subject (graphs from different animals treated as distinct modalities).

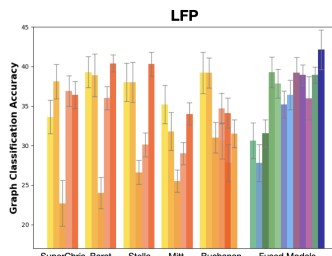 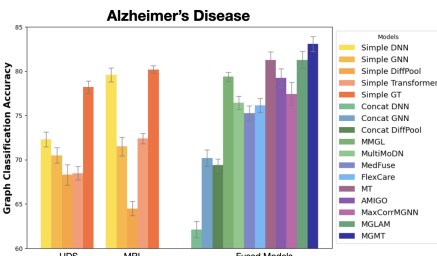

Figure 4: Test accuracies of baseline models on the LFP and Alzheimer's disease datasets. Each bar represents the average test accuracy across 5 folds, along with the corresponding standard error. In both applications, MGMT consistently outperforms all other models.

**Baseline comparisons.** We compare against four families of baselines: (1) *single-source models* (i.e., trained on each source) such as DNNs (LeCun et al., 2015), GNNs, DiffPool (Ying et al., 2018), Transformers, and Graph Transformers (GAT backbone); (2) *early fusion models:* Concatenated Features using DNN/GNN/DiffPool feature extractors (Ngiam et al., 2011; Baltrušaitis et al., 2018; Lau et al., 2019); (3) general-purpose multimodal fusion frameworks such as MMGL (Zheng et al., 2022), MultiMoDN (Swamy et al., 2023), FlexCare (Xu et al., 2024), MedFuse (Hayat et al., 2022), and Meta-Transformer (Ma et al., 2022), and (4) recent multi-graph learning methods tailored to the multi-graph-to-label setting, including AMIGO (Nakhli et al., 2023), MaxCorrMGNN (D'Souza et al., 2023), and MGLAM (Fu et al.), which operate on multiple graphs per entity via shared contexts, correlation-based objectives, or bag-of-graphs pooling (see Appendix A6 for implementation details).

**Ablation studies.** We quantify the contribution of each component by (1) removing adaptive depth selection (i.e., using the final Transformer layer); (2) removing supernode selection (i.e., including all nodes in the meta-graph); (3) removing inter-graph edges; (4) removing intra-graph edges; and (5) disabling both the meta-graph and adaptive depth (i.e., late fusion of fixed-depth Transformer outputs). Results are presented in Tables A2, A3 and Figures 3, 4, 2.

## 5.1 Experimental Setup

**Architecture and training.** Across datasets, MGMT uses `TransformerConv` layers with global max or mean pooling to form graph-level embeddings. Models are trained on 80% of the data, with 10% for validation and 10% for testing, using Adam and early stopping on validation loss. For real datasets, we use 5-fold cross-validation. Hyperparameters, including number of layers, dropout, learning rate, epochs, and node-importance thresholds, are tuned with Optuna (100 trials), selecting the best configuration by validation performance. For simulation studies, we run 50 independent trials and report mean test accuracy with standard errors.

**Runtime and scalability.** Appendix A9 provides component-wise time complexity, empirical runtime profiling (average per-epoch and stage-wise breakdowns), and controlled scalability experiments over graph size, number of graphs per sample count, sample size, and feature dimensionality, showing practical efficiency and predictable scaling comparable to Transformer-based graph architectures.

## 5.2 Synthetic Experiments

We simulate five graphs per sample under varying feature mechanisms, number of nodes, sample size, and noise. Each node has a $p$-dimensional feature; a subset of nodes is *informative* (their features influence the graph-level binary target), and the rest are *non-informative* noise. We create an intermediate binary label for each modality, then aggregate them into an entity-level label by applying a threshold to a weighted sum of these modality-specific labels. **Experiment 1** (Setting 1; Appendix A8): informative-node features are drawn from modality-specific multivariate Gaussians; labels use a linear thresholding rule; $n = 100$. **Experiment 2** (Setting 2; Appendix A8): features for informative nodes are generated using a Gaussian Process to induce temporal structure across features; labels use a nonlinear function (sinusoidal and quadratic); $n = 100$. **Experiment 3**: follows the same setting as Experiment 2 but increases the graph size and sample size. Each graph has 50 nodes, with 40 designated as important. The sample size is increased to 2,000, allowing us to assess MGMT's performance at scale under complex, multimodal conditions.

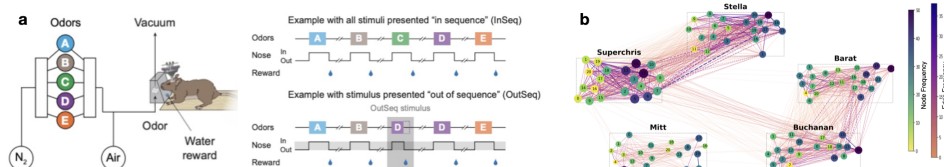

Figure 5: **(a) Neural recordings from CA1 during a sequence memory task.** Rats performed a self-paced odor sequence task, judging each odor in a five-item sequence (A–E) as "in sequence" (InSeq) or "out of sequence" (OutSeq) while odors were delivered through a single port. **(b) Cross-animal supernode and edge frequency map from MGMT.** Each dashed box corresponds to one rat; node size and color indicate supernode selection frequency, and line color reflects edge occurrence frequency. High-frequency supernodes and edges cluster in distal CA1 (right side), with cross-rat superedges primarily linking distal regions across animals. Mitt exhibits weaker connectivity.

Across all experiments, MGMT outperforms baseline models (Fig. 3). Ablations show accuracy drops when removing adaptive depth, or inter-graph edges, and degrade most when both meta-graph and adaptive depth are disabled, supporting importance of hierarchical reasoning (Table A3 and Fig. 2).

## 5.3 NEUROSCIENCE APPLICATIONS

### 5.3.1 LOCAL FIELD POTENTIAL (LFP) ACTIVITY IN A SEQUENTIAL MEMORY TASK

We apply MGMT to a challenging problem: predicting the stimulus presented on a given trial using LFP activity from hippocampus (Fig. 5). In this experiment, 5 subjects (rats named SuperChris, Barat, Stella, Mitt, and Buchanan) received repeated presentations of a sequence of stimuli (odors A, B, C, D, or E) at a single odor port and were required to identify whether each stimulus was presented in correct or incorrect sequence position to receive reward. Neural LFP activity was recorded from the dorsal CA1 subregion of the hippocampus as they performed the task (Allen et al., 2016; Shahbaba et al., 2022). Treating each rat as a distinct graph, MGMT borrows power across subjects and fuses subject-specific representations to decode stimulus identity on each trial from LFP.

Each trial is associated with one shared stimulus label (A,B,C,D or E). We construct a separate graph for each rat per trial using its own electrode-level LFP signals. Nodes represent electrodes (vary in number and identity across subjects), and edges capture intra-subject correlations. We then link "supernodes" across rats when their latent embeddings are similar under MGMT's localized attention. Superedges are aligning comparable brain dynamics across animals, effectively "borrowing statistical strength" across rats to reduce noise and stabilize the trial-level representation used for decoding. This is not meant to just simply connect various brain regions across rats, rather alignment of their brain dynamics to strengthen the overall signals by properly borrowing power across rats.

As shown in Table A2, MGMT achieves the highest accuracy (**42.13%** ± 2.52) predicting which odor (A–E) was presented on each trial using the LFP dataset, outperforming all baseline and fusion models. The best competing method, MMGL, reaches 39.28%, with other recent approaches such as MGLAM (38.93%), AMIGO (38.92%), MT (39.20%), MultiMoDN (37.82%), and FlexCare (36.42%) trailing behind. Traditional concatenation-based approaches like DNN and GNN yield substantially lower performance, highlighting the difficulty of this cross-rat decoding task. To our knowledge, these results provide the first direct evidence that the stimulus presented on a given trial can be accurately predicted based on hippocampal LFP activity alone, which highlights the potential of graph data integration approaches in general and the potential of the MGMT model specifically.

Ablation results (Fig. 2) confirm that each architectural component contributes meaningfully to MGMT's performance, with the full model achieving the highest accuracy across all datasets.

**Results of interpretation component.** From a neuroscience perspective, first, we found that informative electrodes clustered on the right side of the electrode array (Fig. 5b). Specifically, highest-frequency supernodes and strongest within-subject connections were consistently concentrated on the right side, and pattern was consistent across subjects. This specific clustering makes sense given that the two electrode arrays targeted different segments of CA1 region: electrodes on the right targeted the distal segment, electrodes on the left the proximal region. The distal segment, where most informative electrodes are located, is more strongly associated with non-spatial inputs (e.g., odors in our case)

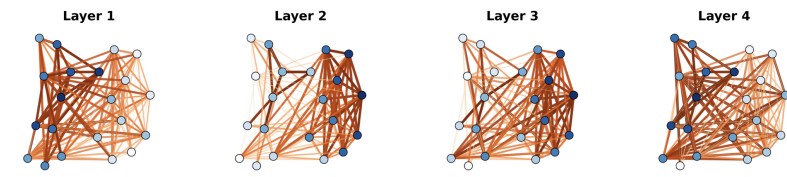

Confidence score $\Gamma_1$ = 0.050    Confidence score $\Gamma_2$ = 0.252    Confidence score $\Gamma_3$ = 0.375    Confidence score $\Gamma_4$ = 0.181

Figure 6: **Layer-wise attention patterns for the LFP data (SuperChris).** Each panel shows the same subject-level LFP connectivity graph, along with the learned depth-confidence scores $\Gamma_\ell$ for each Transformer layer $\ell$, as well as the corresponding edge-level attention scores and node-wise summed attention weights (with warmer colors indicating higher attention or summed weights).

and the proximal segment with visuospatial inputs. Such clustering of informative electrodes in distal CA1 is also consistent with previous work focusing on a different type of non-spatial trial classification (InSeq vs OutSeq (Zhou et al., 2024)). Second, there were interesting variations in the pattern of informative edges across subjects. Although they showed a similar pattern of informative nodes, some subjects showed weaker relationships in edges. For example, subject Mitt showed fewer strong within-subject edges and lower-frequency superedges. In summary, the interpretation module highlights subject-level connectivity differences as the key LFP factors driving performance.

**Depth-aware layers and CA1 circuitry.** The depth-aware component provides a complementary view of these patterns. For each rat, we compute layer-wise depth-confidence scores $\gamma_\ell$ and visualize, on the subject-level LFP connectivity graph, the corresponding edge attention and node-summed attention weights. Fig. 6 shows the layers for SuperChris. Layers with the largest $\Gamma_\ell$ values focus attention on edges linking distal CA1 electrodes, and nodes in this region receive the highest summed attention. In contrast, low-confidence layers distribute attention more diffusely. Thus, the model up-weights layers whose connectivity patterns highlight the distal CA1 subnetwork identified as behaviorally informative in Fig. 5, indicating that depth-aware aggregation selectively amplifies meaningful hippocampal circuitry rather than simply averaging multi-layer embeddings.

### 5.3.2 ALZHEIMER'S DISEASE DETECTION

As an example of broader biomedical applications, we used MGMT for Alzheimer's disease (AD) detection using the data obtained from the National Alzheimer's Coordinating Center (NACC), which standardizes data collected across 46 Alzheimer's Disease Research Centers (ADRCs) in the United States (Beekly et al., 2004; Weintraub et al., 2009). The cohort comprises 1,237 subjects (61.5% HC and 38.5% MCI/AD) with both clinical assessments from the Uniform Data Set (UDS) and structural MRI available. Our goal is to separate subjects with mild cognitive impairment (MCI) or dementia due to Alzheimer's disease from healthy controls (HC).

Following our terminology, this setting is *multi-modal* since each subject is measured via distinct data sources (e.g., MRI vs. clinical assessments) that inhabit different feature spaces and sensing processes. As shown in Fig. 4, the MGMT model consistently outperformed both single-source and baseline fusion models. Moreover, ablations in Fig. 2 show that intra-graph structure and the meta-graph are critical: removing intra-graph edges collapses performance (62.4% vs. 83.1%), removing the meta-graph lowers accuracy to 70.1%, while dropping inter-graph edges (76.5%), supernode selection (78.2%), or adaptive depth (81.2%) yields progressively smaller but consistent declines.

## 6 CONCLUSION, LIMITATIONS, AND FUTURE WORK

We proposed MGMT, a multi-graph learning framework that integrates graph-specific GT encoders with a meta-graph constructed over learned supernodes and superedges, supported by an adaptive depth-aware mechanism for aggregating hierarchical representations. Using both synthetic and real datasets, we showed that MGMT improves accuracy and interpretability over state-of-the-art fusion methods. The framework could be further extended to support node classification and link prediction, incorporate causal masking and counterfactual attribution for genuinely causal importance estimates (see Appendix A14), and improve computational efficiency (see Appendix A15).

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

## A1 GRAPH TRANSFORMER WITH LOCALIZED GRAPH-AWARE ATTENTION

The standard Transformer architecture employs a global self-attention mechanism in which every token attends to all others. This is computationally inefficient and often inappropriate in the context of graph-structured data, where meaningful interactions are localized to a node's immediate neighborhood. To bridge this gap, we adopt the localized graph-aware attention formulation proposed by Shi et al. (2020), which restricts attention to a node's 1-hop neighbors.

To preserve self-information, we extend the neighborhood to include the node itself. Specifically, we define $\bar{\mathcal{N}}(u) = \mathcal{N}(u) \cup \{u\}$, ensuring each node can incorporate its own features during attention-based message passing.

Let $\boldsymbol{H}^{(\ell-1)} = \{\boldsymbol{H}_1^{(\ell-1)}, \ldots, \boldsymbol{H}_N^{(\ell-1)}\}$ denote the set of node features from the previous layer. Each node $u$ aggregates information from its extended neighborhood $v \in \bar{\mathcal{N}}(u)$ using the following multi-head self-attention mechanism.

For each attention head $m = 1, \ldots, M$ and layer $\ell = 1, \ldots, L$:

**1. Linear Projections (queries, keys, values):**

$$\boldsymbol{Q}_u^{(\ell,m)} = \boldsymbol{W}_Q^{(\ell,m)} \boldsymbol{h}_u^{(\ell-1)} + \boldsymbol{b}_Q^{(\ell,m)}, \tag{A10}$$

$$\boldsymbol{K}_v^{(\ell,m)} = \boldsymbol{W}_K^{(\ell,m)} \boldsymbol{h}_v^{(\ell-1)} + \boldsymbol{b}_K^{(\ell,m)}, \tag{A11}$$

$$\boldsymbol{V}_v^{(\ell,m)} = \boldsymbol{W}_V^{(\ell,m)} \boldsymbol{h}_v^{(\ell-1)} + \boldsymbol{b}_V^{(\ell,m)}. \tag{A12}$$

The learnable matrices $\boldsymbol{W}_Q^{(\ell,m)}$, $\boldsymbol{W}_K^{(\ell,m)}$, and $\boldsymbol{W}_V^{(\ell,m)}$ are referred to as the *Query*, *Key*, and *Value* projection matrices, respectively. These matrices project each node's feature vector into three distinct spaces:

- The **Query** vector $\boldsymbol{Q}_u^{(\ell,m)}$ represents the type of information that node $u$ seeks from its neighbors.

- The **Key** vector $\boldsymbol{K}_v^{(\ell,m)}$ encodes what information neighbor node $v$ can provide.

- The **Value** vector $\boldsymbol{V}_v^{(\ell,m)}$ contains the actual content to be aggregated.

This separation allows the model to compute a relevance score between nodes before deciding how much information to share.

**2. Attention Score Calculation:** The attention coefficient from node $u$ to neighbor $v \in \bar{\mathcal{N}}(u)$ is computed as:

$$\alpha_{uv}^{(\ell,m)} = \frac{\exp\left(\frac{\boldsymbol{Q}_u^{(\ell,m)\top} \boldsymbol{K}_v^{(\ell,m)}}{\sqrt{d_h}}\right)}{\sum_{r \in \bar{\mathcal{N}}(u)} \exp\left(\frac{\boldsymbol{Q}_u^{(\ell,m)\top} \boldsymbol{K}_r^{(\ell,m)}}{\sqrt{d_h}}\right)}, \tag{A13}$$

where $d_h$ is the dimensionality of each head.

**3. Neighborhood Aggregation:**

$$\boldsymbol{Z}_u^{(\ell,m)} = \sum_{v \in \bar{\mathcal{N}}(u)} \alpha_{uv}^{(\ell,m)} \boldsymbol{V}_v^{(\ell,m)}. \tag{A14}$$

**4. Multi-Head Output and Update:** The outputs from all heads are concatenated and linearly transformed:

$$\hat{\boldsymbol{H}}_u^{(\ell)} = \boldsymbol{W}_O^{(\ell)} \left[\boldsymbol{Z}_u^{(\ell,1)} \| \cdots \| \boldsymbol{Z}_u^{(\ell,M)}\right] + \boldsymbol{b}_O^{(\ell)}, \tag{A15}$$

where $\|$ denotes concatenation across heads, and $\boldsymbol{W}_O^{(\ell)} \in \mathbb{R}^{d \times d}$, $\boldsymbol{b}_O^{(\ell)} \in \mathbb{R}^d$ are learnable projections.

This formulation allows each node to dynamically attend to its extended local neighborhood, learning rich contextual representations while respecting the sparse structure of the input graph. The learned attention scores can also be used for interpretability and identifying important nodes and edges, as discussed in subsection 3.2.

## A2 DEPTH-AWARE AGGREGATION IN MGMT

To enhance the robustness of graph-specific representation learning and mitigate sensitivity to the choice of Transformer depth, we introduce an adaptive depth-aware fusion strategy inspired by recent developments in graph learning Zhou et al. (2024). Rather than relying on a fixed-depth stack, we aggregate node embeddings across multiple Transformer layers, weighted by their contribution to graph-level prediction performance.

Let $\boldsymbol{H}_{ik}^{(\ell)} \in \mathbb{R}^{N_i \times d}$ denote the node embeddings of graph $i$ in samples (instances) $k$ after the $\ell$-th Graph Transformer layer, for $\ell = 1, \ldots, L$, $i = 1, \ldots, n$ and $k = 1, \ldots, K$. Here, $K$ is the total number of samples, and $n$ is the number of graphs per sample. To evaluate the representational quality of each layer, we compute a graph-level representation by applying mean pooling over the node embeddings:

$$\bar{\boldsymbol{H}}_{ik}^{(\ell)} = \frac{1}{N_i} \mathbf{1}_{N_i}^{\top} \boldsymbol{H}_{ik}^{(\ell)} \in \mathbb{R}^{1 \times d}. \tag{A16}$$

Each pooled graph embedding $\bar{\boldsymbol{H}}_{ik}^{(\ell)}$ is passed through a lightweight classifier to obtain predictions, and its predictive quality is evaluated using the graph-level label. Let $Y_k \in \{1, \ldots, |\mathcal{Y}|\}$ be the true label for sample $k$. The classification error for graph $i$ at depth $\ell$ is computed as:

$$\epsilon_i^{(\ell)} = \frac{\sum_{k=1}^{K} \beta_{ik}^{(\ell)} \Vdash \left\{ Y_k \neq \arg\max_y \operatorname{softmax}\left(\bar{\boldsymbol{H}}_{ik}^{(\ell)}\right) \right\}}{\sum_{k=1}^{K} \beta_{ik}^{(\ell)}} \tag{A17}$$

where $\beta_{ik}^{(\ell)}$ is the weight assigned to graph $i$ in sample $k$ at depth $\ell$.

The confidence score for the $\ell$-th layer of graph $i$ is defined as:

$$\Gamma_i^{(\ell)} = \frac{1}{2} \log\left(\frac{1 - \epsilon_i^{(\ell)}}{\epsilon_i^{(\ell)}}\right). \tag{A18}$$

To emphasize misclassified samples, sample weights are updated between depths using:

$$\beta_{ik}^{(\ell+1)} \propto \beta_{ik}^{(\ell)} \exp\left(\Vdash \left\{ Y_k \neq \arg\max_Y \operatorname{softmax}\left(\bar{\boldsymbol{H}}_{ik}^{(\ell)}\right) \right\} \cdot \Gamma_i^{(\ell)}\right). \tag{A19}$$

The confidence scores $\Gamma_i^{(\ell)}$ are used to weight both the depth-wise fused node embeddings and the attention scores across Transformer layers, ensuring that layers contributing most to prediction are emphasized during supernode extraction and representation learning.

## A3 MATHEMATICAL PROOFS

*Proof of Theorem 4.3.* For simplicity, we omit graph-specific subscripts throughout the proof (e.g. $\boldsymbol{X}$ instead of $\boldsymbol{X}_i$) as the arguments apply universally for all graphs. Consider the Graph Transformer (GT) structure with a single head $m = 1$. For each layer $\ell = 1, \ldots, L$, let $\boldsymbol{W}_Q^{(\ell)} = \boldsymbol{W}_K^{(\ell)} = \mathbf{0}$, $\boldsymbol{W}_V^{(\ell)} = \boldsymbol{I}$, and $\boldsymbol{b}_V^{(\ell)} = \mathbf{0}$ in equation 1. Here $\boldsymbol{I}$ is the identity matrix and $\mathbf{0}$ denotes matrix/vector of all zeros. For the feedforward layer in equation 2, set weights as $\boldsymbol{I}$, bias as $\mathbf{0}$, and remove the residual connection and normalization layer. Then for each edge $(u, v) \in \mathcal{E} \cup \{(u, u)\}$, the updating rules in

equation 1 and equation 2 simplifies to

$$\boldsymbol{Q}_u^{(\ell)} = \boldsymbol{b}_Q^{(\ell)},$$
$$\boldsymbol{K}_v^{(\ell)} = \boldsymbol{b}_K^{(\ell)},$$
$$\boldsymbol{V}_v^{(\ell)} = \boldsymbol{H}_v^{(\ell-1)},$$
$$\alpha_{uv}^{(\ell)} = \frac{\exp\left(\frac{\boldsymbol{Q}_u^{(\ell)\top}\boldsymbol{K}_v^{(\ell)}}{\sqrt{d}}\right)}{\sum_{v' \in \bar{\mathcal{N}}(u)} \exp\left(\frac{\boldsymbol{Q}_u^{(\ell)\top}\boldsymbol{K}_{v'}^{(\ell)}}{\sqrt{d}}\right)},$$
$$\boldsymbol{H}_u^{(\ell)} = \sigma\left(\sum_{v \in \mathcal{N}(u)} \alpha_{uv}^{(\ell)} \boldsymbol{V}_v^{(\ell)}\right).$$

It is clear that the attention matrix $\boldsymbol{\alpha}^{(\ell)}$ reduces to $\mathcal{M}(\boldsymbol{A}) = \text{softmax}(\boldsymbol{A} + \boldsymbol{I})$. Recall that the initial embedding $\boldsymbol{H}^{(0)} = \boldsymbol{X}$, we can explicitly expand the recursive updating rule above, and write the embeddings for each layer $\ell$ in the following compact form:

$$\boldsymbol{H}^{(\ell)} = \mathcal{U}^\ell(\boldsymbol{X}; \mathcal{M}(\boldsymbol{A}), \sigma).$$

Let $\Gamma^{(\ell)} = \eta_\ell$, for $\ell = 1, \ldots, L$ in equation 3, the graph-specific fused embeddings can be represented as

$$\sum_{\ell=1}^{L} \eta_\ell \cdot \mathcal{U}^\ell(\boldsymbol{X}; \mathcal{M}(\boldsymbol{A}), \sigma),$$

which satisfies Definition 4.1 with identity mapping $f(\cdot)$. $\qquad\square$

**Remark A1.** *While the depth-aware fusion step in equation 3 is highly flexible and can accommodate any set of weights $\{\Gamma_\ell\}_{\ell=1}^{L}$, we employ the confidence score weights defined in Section A2 to adaptively aggregate the latent representations that yield the highest classification accuracy.*

*Proof of Theorem 4.4.* Similar to the proof of Theorem A2, we will show $\mathcal{F}_{\text{late}} \subseteq \mathcal{F}_M$ and the desired results follow directly from the definition of approximation error in equation 9.

Consider a class of pooling functions that concatenates the graph-specific pooled embeddings, formally,

$$\text{ConcatPool}(\boldsymbol{H}_M^{(0)}) = \Big\|_{i=1}^{n} \text{Pool}_{\mathcal{S}_i}(\boldsymbol{H}_M^{(0)}), \tag{A20}$$

where $\|$ denotes the concatenation operation, $\text{Pool}_{\mathcal{S}_i}(\cdot) : \mathbb{R}^{|\mathcal{S}_M| \times d} \mapsto \mathbb{R}^{h'}$, as defined in equation A24, is the global pooling function restricted to $\mathcal{S}_i$. Hence $\text{ConcatPool}(\boldsymbol{H}_M^{(0)}) : \mathbb{R}^{|\mathcal{S}_M| \times d} \mapsto \mathbb{R}^{nh'}$ represents the concatenation of graph-specific embeddings.

Further, let $D(\{\boldsymbol{W}_{\text{MLP},i}^{(1)}\}_{i=1}^{n})$ be the diagonal block matrix with diagonal elements $\{\boldsymbol{W}_{\text{MLP},i}^{(1)}\}_{i=1}^{n}$, then one can easily check that equation A20 can be rewritten as

$$\begin{aligned}
\mathcal{F}_{\text{late}} = \Big\{ f : \mathbb{R}^{|\mathcal{S}_M| \times d} &\mapsto \mathbb{R}^{|\mathcal{Y}|} \,\Big|\, f = \boldsymbol{W}_{\text{MLP}}^{(2)} \sigma\Big(\boldsymbol{W}_{\text{MLP}}^{(1)} \text{Pool}(\text{GT}(\boldsymbol{H}_M^{(0)}))\Big), \\
&\gamma > 1, \boldsymbol{W}_V = \boldsymbol{I}, \boldsymbol{b}_V = \boldsymbol{0}, \\
&\text{Pool}(\cdot) = \text{ConcatPool}(\cdot), \\
&\boldsymbol{W}_{\text{MLP}}^{(1)} = D(\{\boldsymbol{W}_{\text{MLP},i}^{(1)}\}_{i=1}^{n}), \\
&\boldsymbol{W}_{\text{MLP}}^{(2)} = w_1 \boldsymbol{W}_{\text{MLP},1}^{(2)} \| \cdots \| w_n \boldsymbol{W}_{\text{MLP},n}^{(2)} \Big\},
\end{aligned} \tag{A21}$$

where $\gamma, \boldsymbol{W}_V, \boldsymbol{b}_V$ are parameters of the Graph Transformer layer as defined in equation A25. Finally, from equation 8 and equation A21, it is clear that $\mathcal{F}_{\text{late}} \subseteq \mathcal{F}_M$, which concludes the proof. $\qquad\square$

# A4 ADDITIONAL THEORETICAL RESULTS

## A4.1 ADDITIONAL INTRA-GRAPH RESULTS

**Theorem A1.** *Let $\mathcal{M}(\boldsymbol{A}) = \mathrm{softmax}(\boldsymbol{A} + \boldsymbol{I})$ as in Theorem 4.3, the vanilla Graph Transformer is **not** capable of representing L-hop neighborhood mixing.*

*Proof.* Following a similar strategy in Abu-El-Haija et al. Abu-El-Haija et al. (2019), it suffices to show that the vanilla Graph Transformer (GT) fails to represent 2-hop mixing, which in turn implies the inability to represent the general $L$-hop mixing. Consider the particular case, where $m = 1$, $\sigma(x) = x$. As reviewed in Section A1, the final graph embedding of a vanilla GT with depth $L$ can be represented as

$$\boldsymbol{H}^{(L)} = \left[ \prod_{\ell=1}^{L} \mathrm{softmax}\left( (\boldsymbol{A} + \boldsymbol{I}) \odot \boldsymbol{\alpha}^{(\ell)} \right) \right] \boldsymbol{X} \prod_{\ell=1}^{L} \boldsymbol{W}_V^{(\ell)},$$

for attention matrices $\{\boldsymbol{\alpha}^{(\ell)}\}_{\ell=1}^{L}$ and weights $\{\boldsymbol{W}_V^{(\ell)}\}_{\ell=1}^{L}$. Here $\odot$ denote the Hadamard product. Let $\boldsymbol{W}^* = \prod_{\ell=1}^{L} \boldsymbol{W}_V^{(\ell)}$, and consider the case where $\eta_1 = 1$ and $\eta_2 = -1$. If the vanilla GT is able to represent 2-hop mixing, there exists an injective mapping $f$ and a configuration of the parameters such that

$$\left[ \prod_{\ell=1}^{L} \mathrm{softmax}\left( (\boldsymbol{A} + \boldsymbol{I}) \odot \boldsymbol{\alpha}^{(\ell)} \right) \right] \boldsymbol{X} \boldsymbol{W}^* = f(\mathcal{M}(\boldsymbol{A})\boldsymbol{X} - \mathcal{M}^2(\boldsymbol{A})\boldsymbol{X}) \tag{A22}$$

holds for any adjacency matrices $\boldsymbol{A}$ and node features $\boldsymbol{X}$.

Consider a fully disconnected graph with $\boldsymbol{A} = \boldsymbol{0}$ and $\boldsymbol{X}$, then $\mathcal{M}(\boldsymbol{A}) = \mathrm{softmax}(\boldsymbol{I}) = \boldsymbol{I}$, and $\mathrm{softmax}\left( (\boldsymbol{A} + \boldsymbol{I}) \odot \boldsymbol{\alpha}^{(\ell)} \right) = \boldsymbol{I}$ for $\ell = 1, \ldots, L$, which implies $\boldsymbol{W}^* = f(\boldsymbol{0})$. On the other hand, consider a graph with a single edge between node 1 and 2, namely, $A_{12} = A_{21} = 1$ and 0 otherwise. Then

$$\mathcal{M}(\boldsymbol{A}) = \underbrace{\begin{bmatrix} 0.5 & 0.5 & 0 & \cdots & 0 \\ 0.5 & 0.5 & 0 & \cdots & 0 \\ 0 & 0 & 1 & \cdots & 0 \\ \vdots & \vdots & \vdots & \ddots & \vdots \\ 0 & 0 & 0 & \cdots & 1 \end{bmatrix}}_{:= \boldsymbol{A}^*}$$

Let $\boldsymbol{X} = \boldsymbol{A}^*$, then $f(\mathcal{M}(\boldsymbol{A})\boldsymbol{X} - \mathcal{M}^2(\boldsymbol{A})\boldsymbol{X}) = f(\boldsymbol{0})$. Furthermore, it is easy to check that

$$\prod_{\ell=1}^{L} \mathrm{softmax}\left( (\boldsymbol{A} + \boldsymbol{I}) \odot \boldsymbol{\alpha}^{(\ell)} \right) = \boldsymbol{A}^*,$$

since features of node 1 and 2 are identical. It follows that $\boldsymbol{A}^* \boldsymbol{W}^* = f(\boldsymbol{0})$.

Combining the two scenarios, we must have $(\boldsymbol{I} - \boldsymbol{A}^*)\boldsymbol{W}^* = \boldsymbol{0}$, which implies that $\boldsymbol{W}_1^* = \boldsymbol{W}_2^*$, where $\boldsymbol{W}_i^*$ is the $i$-th row of $\boldsymbol{W}^*$. Since the choice of node 1 and 2 was arbitrary, all rows of $\boldsymbol{W}^*$ should be identical, hence $\mathrm{rank}(\boldsymbol{W}^*) \leq 1$ and $\mathrm{rank}([\prod_{\ell=1}^{L} \mathrm{softmax}\left( (\boldsymbol{A} + \boldsymbol{I}) \odot \boldsymbol{\alpha}^{(\ell)} \right)] \boldsymbol{X} \boldsymbol{W}^*) \leq 1$, which means the output of $f$ should be at most rank 1 matrices by the equivalence assumption in equation A22. Hence, $f$ cannot be injective, which concludes the proof by contradiction. $\square$

## A4.2 ADDITIONAL INTER-GRAPH RESULTS

Let $\boldsymbol{H}_{\mathcal{S}_i} = \{\boldsymbol{H}_{i,u}\}_{u \in \mathcal{S}_i}$ be the embeddings for supernodes in $\mathcal{S}_i$. Single-graph classifiers that operates on $\boldsymbol{H}_{\mathcal{S}_i}$ can be expressed as

$$\mathcal{F}_i = \left\{ f : \mathbb{R}^{|\mathcal{S}_i| \times d} \mapsto \mathbb{R}^{|\mathcal{Y}|} \; \middle| \; f = \boldsymbol{W}_{\mathrm{MLP}}^{(2)} \sigma\left( \boldsymbol{W}_{\mathrm{MLP}}^{(1)} \mathrm{Pool}(\boldsymbol{H}_{\mathcal{S}_i}) \right) \right\}. \tag{A23}$$

Assume latent representations of the meta graph follow $(\boldsymbol{H}_M^{(0)}, Y) \sim \mathcal{P}_M$, and $(\boldsymbol{H}_{\mathcal{S}_i}, Y) \sim \mathcal{P}_i$ where $\mathcal{P}_i$ is the marginal distribution of $\mathcal{P}_M$ restricted to $\mathcal{S}_i$. The next result shows MGMT achieves smaller approximation error by leveraging information across all graphs.

**Proposition A2.** *Denote the approximation error of MGMT on the meta-graph as $\epsilon(\mathcal{F}_M; \mathcal{P}_M, \mathcal{L})$, and the approximation error of graph-specific classifiers on the sub-graph as $\epsilon(\mathcal{F}_i; \mathcal{P}_i, \mathcal{L})$, then*

$$\epsilon(\mathcal{F}_M; \mathcal{P}_M, \mathcal{L}) \leq \epsilon(\mathcal{F}_i; \mathcal{P}_i, \mathcal{L}).$$

*Proof of Proposition A2.* Without loss of generality, we focus on the cases where both MGMT and graph-specific classifiers has $L_{\mathrm{MLP}} = 2$ layers of MLP and MGMT has $L_{\mathrm{GT}} = 1$ layer of Graph Transformer as specified in equation 8 and equation A23. The same argument below applies to any number of $L_{\mathrm{MLP}}$ and $L_{\mathrm{GT}}$.

First, consider the function class that operates on the meta-graph but only utilizes the nodes from graph $i$, namely,

$$\bar{\mathcal{F}}_i = \left\{ f : \mathbb{R}^{|\mathcal{S}_M| \times d} \mapsto \mathbb{R}^{|\mathcal{Y}|} \,\middle|\, f = \boldsymbol{W}_{\mathrm{MLP}}^{(2)} \sigma\left(\boldsymbol{W}_{\mathrm{MLP}}^{(1)} \mathrm{Pool}_{\mathcal{S}_i}(\boldsymbol{H}_M^{(0)})\right) \right\}, \tag{A24}$$

where $\mathrm{Pool}_{\mathcal{S}_i}$ denote the global pooling operation that restricts on the nodes in $\mathcal{S}_i$. Since

$$\mathrm{Pool}_{\mathcal{S}_i}(\boldsymbol{H}_M^{(0)}) = \mathrm{Pool}(\boldsymbol{H}_{\mathcal{S}_i}),$$

We have that

$$R\left(\boldsymbol{W}_{\mathrm{MLP}}^{(2)} \sigma\left(\boldsymbol{W}_{\mathrm{MLP}}^{(1)} \mathrm{Pool}_{\mathcal{S}_i}(\boldsymbol{H}_M^{(0)})\right); \mathcal{P}_M, \mathcal{L}\right) = R\left(\boldsymbol{W}_{\mathrm{MLP}}^{(2)} \sigma\left(\boldsymbol{W}_{\mathrm{MLP}}^{(1)} \mathrm{Pool}(\boldsymbol{H}_{\mathcal{S}_i})\right); \mathcal{P}_i, \mathcal{L}\right).$$

It follows that

$$\epsilon(\bar{\mathcal{F}}_i; \mathcal{P}_M, \mathcal{L}) = \epsilon(\mathcal{F}_i; \mathcal{P}_i, \mathcal{L}).$$

We claim that $\bar{\mathcal{F}}_i \subseteq \mathcal{F}_M$, and by definition of approximation error in equation 9,

$$\epsilon(\mathcal{F}_M; \mathcal{P}_M, \mathcal{L}) \leq \epsilon(\bar{\mathcal{F}}_i; \mathcal{P}_M, \mathcal{L}) = \epsilon(\mathcal{F}_i; \mathcal{P}_i, \mathcal{L}).$$

It remains to show the function class inclusion. Note that we can rewrite $\bar{\mathcal{F}}_i$ as

$$\bar{\mathcal{F}}_i = \left\{ f : \mathbb{R}^{|\mathcal{S}_M| \times d} \mapsto \mathbb{R}^{|\mathcal{Y}|} \,\middle|\, f = \boldsymbol{W}_{\mathrm{MLP}}^{(2)} \sigma\left(\boldsymbol{W}_{\mathrm{MLP}}^{(1)} \mathrm{Pool}_{\mathcal{S}_i}(\mathrm{GT}(\boldsymbol{H}_M^{(0)}))\right), \right.$$
$$\left. \gamma > 1, \boldsymbol{W}_V = \boldsymbol{I}, \boldsymbol{b}_V = \boldsymbol{0} \right\}, \tag{A25}$$

where $\gamma$ is the threshold defined in Section 3.1.3 that determines the connectivity between nodes in the meta-graph, $\boldsymbol{W}_V, \boldsymbol{b}_V$ are parameters for values in the Graph Transformer layer. Setting $\gamma > 1$ results in a fully disconnected meta-graph and together with $\boldsymbol{W}_V = \boldsymbol{I}, \boldsymbol{b}_V = \boldsymbol{0}$, the Graph Transformer layer $\mathrm{GT}(\cdot)$ reduces to an identity mapping, which establishes the equivalence in equation A25.

Finally, from equation 8 and equation A25, it is clear that $\bar{\mathcal{F}}_i \subseteq \mathcal{F}_M$, which concludes the proof. $\quad\square$

### A4.3 L-HOP MIXING VERSUS WEISFEILER-LEMAN

A natural question arises regarding the relationship between L-hop mixing (Theorem 4.3) and Weisfeiler-Leman (WL) expressivity: does the ability to represent L-hop mixing translate into enhanced distinguishing power in the Weisfeiler-Leman test. In this section, we clarify that these are distinct characterizations of model power and provide a formal analysis of MGMT's WL expressivity.

L-hop mixing and WL expressivity measure different aspects of model capability. WL expressivity characterizes *distinguishing power*: whether a model can distinguish non-isomorphic graphs. The 1-dimensional WL test (1-WL) iteratively refines node colorings based on local neighborhood structures, and it is well-established that standard message-passing GNNs are at most as powerful as 1-WL Morris et al. (2019); Jegelka (2022). In contrast, L-hop mixing characterizes *approximation quality*: whether a model can exactly recover target functions that depend on mixed-depth neighborhood information (Theorem 4.1). MGMT's capability of representing L-hop mixing comes from the depth-aware aggregation in equation 1–equation 3, independent of the GT backbone, while MGMT's WL expressivity depends on the GT backbone choice. The empirical results in Table A6 demonstrate that depth-aware aggregation enhances performance regardless of the GT backbone, confirming that L-hop mixing and WL expressivity are complementary properties that jointly contribute to model capability.

**MGMT with Graph Attention Networks (GAT) backbone is 1-WL bounded.** We now formally analyze MGMT's distinguishing power. We adopt notation from Morris et al. (2019); Jegelka (2022). A node coloring $l : \mathcal{V}(\mathcal{G}) \mapsto \Sigma$ maps node $v \in \mathcal{V}(\mathcal{G})$ to color $l(v) \in \Sigma$. A labeled graph $(\mathcal{G}, l)$ is graph $\mathcal{G}$ with node coloring $l : \mathcal{V}(\mathcal{G}) \mapsto \Sigma$. Node coloring $c$ refines coloring $d$, written $c \sqsubseteq d$, if $c(v) = c(w)$ implies $d(v) = d(w)$ for every $v, w \in \mathcal{V}(\mathcal{G})$. Two colorings are equivalent, written $c \equiv d$, if $c \sqsubseteq d$ and $d \sqsubseteq c$. The notation $\{\{\dots\}\}$ denotes a multiset.

For labeled graph $(\mathcal{G}, l)$, 1-WL computes node coloring $c_l^{(t)} : \mathcal{V}(\mathcal{G}) \mapsto \Sigma$ iteratively for $t \geq 0$. Let $c_l^{(0)} = l$ and for each $u \in \mathcal{V}(\mathcal{G})$ and $t \geq 0$:

$$c_l^{(t)}(v) = \text{HASH}\Big( c_l^{(t-1)}(v), \{\{c_l^{(t-1)}(u) \mid u \in \mathcal{N}(v)\}\}\Big), \tag{A26}$$

where HASH is an injective mapping assigning unique colors to distinct input pairs.

The key difference between 1-WL and MGMT's depth-aware GT (Section 3.1.1) is that the former updates based on colorings of $\{v\} \cup \mathcal{N}(v)$ from the previous iteration, while the latter aggregates outputs across all depths/iterations. However, depth aggregation does not make MGMT more powerful than 1-WL in distinguishing power. To see this, we define a 1-WL variation, 1-WL$^+$, that utilizes all depth information. Let $\tilde{c}_l^{(t)} : \mathcal{V}(\mathcal{G}) \mapsto \Sigma$ be the 1-WL$^+$ coloring with $\tilde{c}_l^{(0)} = l$ and $\tilde{c}_l^{(1)} = c_l^{(1)}$. For $t \geq 1$:

$$\tilde{c}_l^{(t)}(v) = \text{HASH}^{(t)}\Big( c_l^{(1)}(v), \dots, c_l^{(t)}(v)\Big), \tag{A27}$$

where HASH$^{(t)}$ is an injective map assigning colors based on 1-WL outputs across all iterations. Despite this additional step beyond vanilla 1-WL, 1-WL$^+$ provides no additional distinguishing power, as established by the following Lemma.

**Lemma A3.** *Let $(\mathcal{G}, l)$ be a labeled graph. Then for all $t \geq 0$, $c_l^{(t)} \equiv \tilde{c}_l^{(t)}$.*

*Proof.* For any $v, w \in \mathcal{V}(\mathcal{G})$, if $\tilde{c}_l^{(t)}(v) = \tilde{c}_l^{(t)}(w)$, we must have $c_l^{(k)}(v) = c_l^{(k)}(w)$, for all $k = 1, \dots, t$ by injectivity of HASH$^{(t)}$, hence $\tilde{c}_l^{(t)} \sqsubseteq c_l^{(t)}$. On the other hand, if $c_l^{(t)}(v) = c_l^{(t)}(w)$, we have $c_l^{(k)}(v) = c_l^{(k)}(w)$ for all $k = 1, \dots, t-1$ by injectivity of HASH. It follows that $\tilde{c}_l^{(t)}(v) = \tilde{c}_l^{(t)}(w)$ since all inputs are equivalent. Hence, we have $c_l^{(t)} \sqsubseteq \tilde{c}_l^{(t)}$, which concludes the proof. $\square$

Following similar arguments as in Morris et al. (2019); Jegelka (2022), the distinguishing power of MGMT's depth-aware GT is upper-bounded by 1-WL$^+$ (hence 1-WL) and reaches maximal capacity when the attention layers in Equations (1)-(2) (corresponding to HASH) and the depth aggregation in Equation (3) (corresponding to HASH$^{(t)}$) are injective functions.

**Going beyond 1-WL.** However, it is possible to extend MGMT beyond 1-WL expressivity. As detailed in Section A12, MGMT's main contribution is delineating a flexible framework for multi-graph fusion where practitioners can freely replace the GAT backbone with other GT variants suitable for the task, such as Graphormer Ying et al. (2021). As shown in Ying et al. (2021), incorporating structural encodings and global attention leads to strictly more expressive power than the 1-WL test. Therefore, MGMT with Graphormer backbone can technically break the 1-WL limitation discussed in the Lemma above.

## A5 THEORETICAL FOUNDATIONS OF EMBEDDING-SIMILARITY SUPEREDGES

Graph learning typically assumes that connected nodes have similar features or labels; a smoothness (homophily) prior grounded in the observation that many real-world networks connect like entitiesZhou et al. (2003); Rossi et al. (2022). This assumption is often enforced by minimizing the graph Dirichlet energy (GDE, see Definition 2.1), which is the sum of squared feature differences across edges, thereby yielding smooth node embeddings that are harmonic functions on the graph Rossi et al. (2022). A function $f$ is defined as "harmonic" if it satisfies the discrete Laplace equation $\boldsymbol{L}f = 0$, where $\boldsymbol{L} := \boldsymbol{D} - \boldsymbol{A}$ is the combinatorial graph Laplacian (with $\boldsymbol{A}$ as the adjacency matrix and $\boldsymbol{D}$ as the diagonal degree matrix). This condition is *equivalent* to the averaging rule $f(u) = \frac{1}{\deg(u)} \sum_{v \sim u} f(v)$ for every unlabeled node $u$. Minimizing the GDE enforces this harmonic

Table A1: Model Category Summary with Fusion Strategy, Graph Modeling, and Attention Usage

| Category | Model Type | Fusion Method | Novel Model | Graph Structured Modeling | Attention-Based |
|---|---|---|---|---|---|
| Single-Source (No Fusion) | Simple DNN | × | × | × | × |
| | Simple GNN | × | × | ✓ | × |
| | Simple DiffPool | × | × | ✓ | × |
| | Simple Transformer | × | × | × | ✓ |
| | Simple Graph Transformer | × | × | ✓ | ✓ |
| Concatenation Fusion | Concatenated Features (DNN) | ✓ | × | × | × |
| | Concatenated Features (GNN) | ✓ | × | ✓ | × |
| | Concatenated Features (DiffPool) | ✓ | × | ✓ | × |
| Multimodal Fusion Baselines | MMGL Zheng et al. (2022) | ✓ | × | ✓ | ✓ |
| | MultiMoDN Swamy et al. (2023) | ✓ | × | × | × |
| | MedFuse Hayat et al. (2022) | ✓ | × | × | × |
| | FlexCare Xu et al. (2024) | ✓ | × | × | ✓ |
| | Meta-Transformer (MT) Ma et al. (2022) | ✓ | × | × | ✓ |
| MGMT Ablation Variants | MGMT w/o Adaptive Depth Selection | ✓ | ✓ | ✓ | ✓ |
| | MGMT w/o Supernode Selection | ✓ | ✓ | ✓ | ✓ |
| | MGMT w/o Inter-graph Edges | ✓ | ✓ | ✓ | ✓ |
| | MGMT w/o Intra-graph Edges | ✓ | ✓ | ✓ | ✓ |
| | MGMT w/o Meta-Graph and Adaptive Depth | ✓ | ✓ | ✓ | ✓ |
| Proposed Model | MGMT | ✓ | ✓ | ✓ | ✓ |

property, a classical result in graph-based semi-supervised learning Zhu et al. (2003); Zhou et al. (2003). Such smoothness-based regularization has proven beneficial in both classical label propagation and modern GNNs when the assumption holds, as it suppresses noise and aligns learned representations with network structure.

From a spectral viewpoint, GDE minimization penalizes "high-frequency" components (rapid changes across adjacent nodes), and standard message passing performs neighborhood averaging (a low-pass operation), which denoises features while preserving cluster-level structure; this explains the strong empirical performance of label propagation and Graph Convolutional Network (GCN)-style models on homophilous benchmarksShuman et al. (2013); Jiang et al. (2019); Wu et al. (2019); Oono & Suzuki (2019). If the smoothness prior is violated (heterophilic graphs where adjacent nodes differ), aggressive smoothing can blur distinctions and degrade performance Zhu et al. (2020). This "feature mixing" is well documented: on heterophilous graphs, even shallow neighbor-averaging can wash out class signal, and deeper stacks exacerbate *over-smoothing*, where node embeddings become nearly indistinguishableWu et al. (2019); Li et al. (2018). MGMT's design explicitly leverages these principles: it links nodes across graphs only when their latent representations are similar, extending the homophily prior to inter-graph connections. Concretely, by thresholding latent similarity, MGMT restricts message passing to approximately homophilous (low-GDE) superedges, mitigating heterophily-induced feature mixing; this mirrors observations that learning/selecting edges to reduce Dirichlet energy improves downstream accuracy Chen et al. (2020). By keeping cross-graph GDE low, this construction ensures information is shared along feature-consistent (smooth) superedges, thereby bolstering MGMT's empirical performance.

**Definition 2.1 (Graph Dirichlet Energy).** For a graph with adjacency matrix $\boldsymbol{A}$ and node feature matrix $\boldsymbol{X}$, the Dirichlet energy (graph signal smoothness) is defined as

$$\Omega(\boldsymbol{A}, \boldsymbol{X}) = \frac{1}{2n^2} \sum_{i,j} A_{ij} \|x_i - x_j\|^2 = \frac{1}{n^2} \operatorname{tr}(\boldsymbol{X}^\top \boldsymbol{L} \boldsymbol{X}),$$

where $\boldsymbol{L} = \boldsymbol{D} - \boldsymbol{A}$ is the graph Laplacian and $D_{ii} = \sum_j A_{ij}$ Chen et al. (2020). This quantity measures how smoothly the features $\boldsymbol{X}$ vary across the edges. A smaller $\Omega(\boldsymbol{A}, \boldsymbol{X})$ indicates that connected nodes have more similar features.

A trivial minimizer of $\Omega(\boldsymbol{A}, \boldsymbol{X})$ is the disconnected graph with no edges ($\boldsymbol{A} = 0$), yielding the minimum GDE of 0 Chen et al. (2020). However, in practice, one often imposes constraints such as a fixed number of edges, a connectivity requirement, or regularization terms to avoid this degenerate solution. The objective thus becomes to add only the most "homophilous" edges that connect similar nodes, thereby keeping the GDE low. Under this motivation, minimizing $\Omega(\boldsymbol{A}, \boldsymbol{X})$ reduces to selecting the most "homophilous" edges. MGMT implements this principle directly. It computes all pairwise similarities between supernode embeddings and forms superedges only if the similarity surpasses a data-driven threshold automatically selected via cross-validation (detailed in Section 3.1.3). Finally, we note that while the GDE in Definition 2.1 is based on squared feature

Table A2: Accuracy (± standard error) for different models across datasets, grouped by model family.

| Category | Model | Alzheimer | LFP Data | Experiment 1 | Experiment 2 | Experiment 3 |
|---|---|---|---|---|---|---|
| Feature-Concatenation | DNN | $62.13 \pm 0.91$ | $30.62 \pm 2.28$ | $61.87 \pm 2.27$ | $56.10 \pm 1.13$ | $63.74 \pm 0.56$ |
| | GNN | $70.17 \pm 0.93$ | $27.80 \pm 2.34$ | $55.64 \pm 2.36$ | $64.20 \pm 1.20$ | $67.17 \pm 0.60$ |
| | DiffPool | $69.40 \pm 0.70$ | $31.53 \pm 1.76$ | $53.78 \pm 1.75$ | $65.80 \pm 0.89$ | $64.81 \pm 0.44$ |
| General-purpose Multimodal | MMGL | $79.38 \pm 0.52$ | $39.28 \pm 1.93$ | $59.20 \pm 1.04$ | $62.80 \pm 0.84$ | $68.75 \pm 0.12$ |
| | MultiMoDN | $76.44 \pm 0.75$ | $37.82 \pm 1.82$ | $60.40 \pm 1.67$ | $61.50 \pm 1.01$ | $65.10 \pm 0.50$ |
| | MedFuse | $75.27 \pm 0.84$ | $35.17 \pm 1.71$ | $59.70 \pm 1.52$ | $64.35 \pm 0.96$ | $63.84 \pm 0.53$ |
| | FlexCare | $76.14 \pm 0.79$ | $36.42 \pm 1.88$ | $61.10 \pm 1.39$ | $69.82 \pm 0.91$ | $64.03 \pm 0.56$ |
| | MT | $81.29 \pm 0.92$ | $39.20 \pm 2.96$ | $62.31 \pm 1.24$ | $66.30 \pm 1.12$ | $69.24 \pm 0.34$ |
| SOTA Multi-graph | AMIGO | $79.23 \pm 1.04$ | $38.92 \pm 1.32$ | $58.68 \pm 2.12$ | $65.73 \pm 0.89$ | $70.42 \pm 0.79$ |
| | MaxCorrMGNN | $77.43 \pm 1.35$ | $35.97 \pm 2.73$ | $56.23 \pm 1.32$ | $63.72 \pm 1.64$ | $71.46 \pm 1.01$ |
| | MGLAM | $81.29 \pm 0.96$ | $38.93 \pm 1.02$ | $61.96 \pm 1.04$ | $62.29 \pm 1.29$ | $69.52 \pm 0.39$ |
| Proposed model | **MGMT** | $\mathbf{83.11 \pm 0.84}$ | $\mathbf{42.13 \pm 2.52}$ | $\mathbf{65.47 \pm 2.39}$ | $\mathbf{69.90 \pm 1.19}$ | $\mathbf{73.21 \pm 0.59}$ |

differences, we observed in practice that MGMT's performance is not sensitive to the specific choice of similarity metric used for this filtering step (Section A11).

## A6  DETAILED DESCRIPTIONS OF BASELINE MODELS

This appendix details the baselines used to evaluate our method. Table A1 provides a summary comparison of the baseline models.

### A6.1  SINGLE-SOURCE MODELS (NO FUSION)

We assess per-source predictive signal with five baselines: (i) DNN on flattened node features (edges ignored); (ii) a message-passing GNN with graph-convolution layers over the given topology; (iii) DiffPool for hierarchical pooling into coarser clusters Ying et al. (2018); (iv) Transformer over node-feature sequences (no structural encoding); and (v) Graph Transformer that attends over 1-hop neighborhoods to incorporate local structure.

### A6.2  FEATURE-CONCATENATION FUSION MODELS

These models use early fusion: each source is encoded by a source-specific extractor, the resulting embeddings are concatenated, and a shared DNN classifier is applied. Concretely, we consider (i) DNN-fusion with per-source DNN encoders; (ii) GNN-fusion with per-source GCN layers and graph-level pooling prior to concatenation; and (iii) DiffPool-fusion using per-source DiffPool encoders to produce graph-level embeddings that are concatenated and classified by a DNN.

### A6.3  GENERAL-PURPOSE MULTIMODAL FUSION

We benchmark against recent multimodal frameworks with distinct fusion strategies: (i) MMGL Zheng et al. (2022), which learns shared/specific embeddings via modality-aware representation learning and models subject-level similarity with a GNN; (ii) MultiMoDN Swamy et al. (2023), a modular design with independent encoders and late fusion, without structural reasoning; (iii) MedFuse Hayat et al. (2022), which aligns modalities in a shared latent space using contrastive/reconstruction losses, without explicit intra- or inter-modality structure; (iv) FlexCare Xu et al. (2024), which uses modality-specific encoders and a Transformer fusion layer for heterogeneous clinical data, but no graph-based reasoning; and (v) Meta-Transformer (MT) Ma et al. (2022), which uses modality prompts with a shared Transformer over unstructured inputs, without topological modeling. MGMT differs by jointly capturing both intra- and inter-graph relations through an attention-based meta-graph.

Most of these benchmark models were not originally designed for graph-structured inputs (they expect tabular, imaging, or clinical features). To compare fairly, we first converted each graph into a fixed-length vector by running the same graph-specific encoder used in MGMT (TransformerConv with global pooling and adaptive-depth aggregation) and using the resulting graph-level embedding as a "tabular" feature vector. For methods with multi-stream inputs (e.g., MultiMoDN, FlexCare,

MedFuse), we fed one embedding per graph; for single-stream methods (e.g., Meta-Transformer), we concatenated the graph embeddings. All baselines used identical train/val/test splits, per-graph standardization, a learned linear projection to align embedding dimensions when required, and the same Optuna budget for hyperparameter tuning.

### A6.4 MULTI-GRAPH LEARNING MODELS

Finally, we include three recent multi-graph learning methods as baselines that explicitly operate on multiple graphs per entity: (i) AMIGO (Nakhli et al., 2023) is a sparse multi-graph transformer model that processes multiple modality-specific graphs for each subject and uses a shared context mechanism to exchange information across graphs. Each graph is first encoded by a graph transformer to obtain a graph-level representation; AMIGO then employs cross-graph attention between these representations and a shared context token to produce a fused embedding used for prediction; (ii) MaxCorrMGNN (D'Souza et al., 2023) is a multi-graph neural network that encourages the embeddings of different graphs from the same subject to be maximally correlated. It learns graph-level embeddings for each graph and optimizes a correlation-based objective across graphs, followed by a classifier on the fused embedding. This explicitly aligns graph-specific representations while retaining graph-specific encoders; MGLAM (Fu et al.) treats each subject as a "bag of graphs", and learns adaptive weights over graphs within the bag. It first computes graph-level embeddings for each graph, then aggregates them via an attention-like mechanism that learns per-graph importance scores, yielding a subject-level representation for downstream prediction.

In our experiments, all three methods are instantiated on the same graph-specific design as MGMT, using the same per-graph encoders (where applicable), train/validation/test splits, and comparable hyperparameter tuning budgets. Unlike MGML-style multimodal fusion, these methods are designed to handle multiple graphs per subject, but they fuse graphs at the level of graph embeddings or bags of graphs. In contrast, MGMT constructs an explicit meta-graph over *supernodes*, enabling fine-grained cross-graph message passing that preserves intra-graph topology while modeling inter-graph structure.

## A7 ABLATION STUDY

We assess the contribution of each MGMT component through a series of ablations where one or several modules are removed while the rest of the architecture is kept fixed: (i) *w/o Adaptive Depth Selection*: replace confidence-weighted layer aggregation with final-layer-only features, disabling depth-wise ensembling; (ii) *w/o Supernode Selection*: bypass attention-based node filtering so that all nodes enter the meta-graph; (iii) *w/o Inter-graph Edges*: keep only within-graph edges, removing cross-graph interactions in the meta-graph; (iv) *w/o Intra-graph Edges*: keep only cross-graph edges, discarding within-graph structure for the supernodes; (v) *w/o Meta-Graph and Adaptive Depth*: omit the meta-graph entirely, fix encoder depth, and perform early fusion via concatenated pooled graph outputs.

Numerical results for each ablation across datasets are reported in Table A3, with corresponding accuracy plots in Fig. 2. Several consistent patterns emerge.

First, **removing the meta-graph and adaptive depth** leads to the largest degradation on all tasks (e.g., from $83.11\%$ to $70.12\%$ on Alzheimer and from $42.13\%$ to $27.80\%$ on LFP). This variant reduces MGMT to an early-fusion model over pooled graph embeddings, eliminating both subgraph-level cross-graph message passing and the ability to combine information across depths. The sharp drop indicates that the meta-graph is not a cosmetic addition: explicitly modeling interactions between a small set of informative supernodes drawn from the multiple graphs of each entity is crucial for integrating heterogeneous graphs and stabilizing predictions in multi-graph settings.

Second, **disabling adaptive depth** ("w/o Adaptive Depth Selection") consistently hurts performance (e.g., from $83.11\%$ to $81.20\%$ on Alzheimer, and from $42.13$ to $40.64\%$ on LFP). Together with the depth-confidence and attention visualizations in the main paper (Fig. 6), this supports our interpretation of the depth-aware module as more than a simple multi-layer average: layers with high confidence scores focus their attention on behaviorally relevant substructures (e.g., distal CA1 in the LFP dataset), whereas low-confidence layers exhibit more diffuse patterns. When we force the model to use only the final layer, it can no longer adaptively emphasize those depths whose

Table A3: Accuracy (± standard error) for different ablation models across datasets.

| Model | Alzheimer | LFP Data | Experiment 1 | Experiment 2 | Experiment 3 |
|---|---|---|---|---|---|
| MGMT w/o Adaptive Depth Selection | $81.20 \pm 0.85$ | $40.64 \pm 2.23$ | $64.20 \pm 2.40$ | $68.80 \pm 1.17$ | $71.45 \pm 0.57$ |
| MGMT w/o Supernode Selection | $78.25 \pm 0.87$ | $41.07 \pm 2.19$ | $62.11 \pm 2.16$ | $67.30 \pm 1.07$ | $69.31 \pm 0.53$ |
| MGMT w/o Inter-graph Edges | $76.59 \pm 0.88$ | $38.91 \pm 2.14$ | $61.72 \pm 2.25$ | $66.90 \pm 1.21$ | $68.35 \pm 0.51$ |
| MGMT w/o Intra-graph Edges | $62.40 \pm 2.43$ | $39.08 \pm 0.97$ | $63.09 \pm 2.42$ | $66.62 \pm 1.23$ | $66.75 \pm 0.62$ |
| MGMT w/o Meta-Graph and Adaptive Depth | $70.12 \pm 0.93$ | $27.80 \pm 2.34$ | $55.64 \pm 2.36$ | $64.20 \pm 1.20$ | $67.17 \pm 0.60$ |
| MGMT | $\mathbf{83.11 \pm 0.84}$ | $\mathbf{42.13 \pm 2.52}$ | $\mathbf{65.47 \pm 2.39}$ | $\mathbf{69.90 \pm 1.19}$ | $\mathbf{73.21 \pm 0.59}$ |

connectivity patterns are most aligned with the task, leading to more misclassifications in cases that require multi-scale aggregation.

Third, **removing supernode selection** ("w/o Supernode Selection") yields a moderate but consistent drop relative to full MGMT across all datasets. This is consistent with the threshold-sensitivity analysis (Section A10) for the supernode importance score: very low thresholds allow keeping many weakly informative nodes in the meta-graph, making it denser and noisier. In contrast, moderate thresholds strike a balance between retaining salient subgraphs and suppressing noise. The ablation corresponds to the extreme case where all nodes are kept (effectively $\tau \to 0$), and the resulting performance degradation indicates that the sparsity-inducing bottleneck provided by supernode selection is important for denoising and interpretability.

Finally, the **inter- and intra-graph edge ablations** clarify how MGMT exploits structure at two complementary levels. Removing inter-graph edges ("w/o Inter-graph Edges") prevents information from flowing across graphs of the same entity; accuracy drops are noticeable (e.g., from $83.11\%$ to $76.59\%$ on Alzheimer), indicating that cross-graph alignment provides a clear gain on top of strong graph-specific encoders. In contrast, removing intra-graph edges ("w/o Intra-graph Edges") discards the original within-graph topology and forces the model to rely solely on similarity-based links between supernodes from different graphs; this leads to a much larger degradation on real datasets (e.g., Alzheimer accuracy falls to $62.40\%$). This pattern is consistent with our similarity-threshold study in Section A10: when the inter-graph similarity threshold $\gamma$ is too low, the meta-graph becomes overly dense and spurious cross-graph edges blur informative graph-specific structure, whereas very high $\gamma$ removes many genuinely aligned supernodes and under-utilizes cross-graph information. The best performance arises at intermediate $\gamma$ values, where inter-graph edges selectively connect strongly aligned supernodes and, as formalized by our smoothness analysis in Section A5, tend to reduce the Dirichlet energy of the label function on the meta-graph. Taken together, the ablations support the view that MGMT needs both well-structured intra-graph connectivity to encode subject- or modality-specific patterns, and a sparse, similarity-driven set of inter-graph edges to tie together truly corresponding regions across graphs; removing either source of structure degrades performance, with the largest failures occurring when the more informative structure for a given task (typically the intra-graph topology) is removed.

Software implementing the algorithms and data experiments is available online at:
`https://anonymous.4open.science/r/new_submission-33A6`

## A8 DETAILS ON SIMULATION SETTINGS

This section provides detailed descriptions of the synthetic data generation processes used in our simulation studies. We consider two controlled settings designed to evaluate the performance of MGMT under varying conditions of noise, feature dependency, and label complexity. Below, we describe the procedures for *Setting 1*, which uses modality-specific noise and a linear classification rule, and *Setting 2*, which introduces temporal dependencies and nonlinear label generation.

SETTING 1: FEATURE GENERATION WITH MODALITY-SPECIFIC NOISE AND LINEAR CLASSIFICATION RULE

Let each graph (modality in this case) consist of $N$ nodes and $d$ features per node. Define a subset of informative nodes $V_0 \subset \{1, \dots, N\}$ with $|V_0| = N_0 < N$, and let $V_1 = \{1, \dots, N\} \setminus V_0$ denote the non-informative nodes.

For each graph $i = 1, \ldots, n$ within each sample $k = 1, \ldots, K$, with graph-specific noise level $\sigma_i$, node features are generated as follows:

- informative nodes $j \in V_0$ have features $\boldsymbol{x}_j^{(k,i)} \sim \mathcal{N}(\mathbf{0}, \boldsymbol{\Sigma}_i)$, where $\boldsymbol{\Sigma}_i \in \mathbb{R}^{d \times d}$ has ones on the diagonal and off-diagonal entries sampled uniformly from $[-\sigma_i, \sigma_i]$.

- Non-informative nodes $j \in V_1$ have features $\boldsymbol{x}_j^{(k,i)} \sim \text{Unif}(0, 0.5)^d$.

The graph-specific graph-level binary label $y_i^{(k)} \in \{0, 1\}$ is determined by the features of informative nodes:

$$y_i^{(k)} = \mathbb{I}\left(\frac{1}{|V_0|} \sum_{j \in V_0} \sum_{r=1}^{d} x_{j,r}^{(k,i)} + \varepsilon^{(k)} > 0\right), \quad \varepsilon^{(k)} \sim \mathcal{N}(0, 0.1).$$

To enable multimodal fusion, a shared target variable is defined by aggregating graph-specific labels:

$$y_{\text{shared}}^{(k)} = \mathbb{I}\left(\sum_{i=1}^{n} w_i y_i^{(k)} \geq \tau\right),$$

where $w_i \in [0, 1]$ are graph-specific weights summing to one, and $\tau \in [0, 1]$ is a threshold parameter.

SETTING 2: TEMPORAL FEATURE DEPENDENCY VIA GAUSSIAN PROCESS

In this setting, features of informative nodes are generated using a Gaussian Process (GP) to introduce temporal dependency across the $d$ features. For $t = 1, \ldots, d$, let $x_t \sim \text{Unif}(0, 1)$, and define the GP with zero mean and a squared exponential kernel:

$$k(x_t, x_{t'}) = \sigma^2 \exp\left(-\frac{(x_t - x_{t'})^2}{l^2}\right),$$

with length-scale $l = 1$ and variance $\sigma^2 = 1$.

For non-informative nodes, features are also sampled from a GP with the same mean function, but with increased kernel variance $\sigma^2 = 2.5$, thereby injecting greater noise and reducing relevance for the target prediction.

The binary target label is defined using a nonlinear and complex function of the averaged features across informative nodes. Let

$$\boldsymbol{x} = \frac{1}{|V_0|} \sum_{j \in V_0} \boldsymbol{x}_j \in \mathbb{R}^d,$$

and define three projection vectors $\boldsymbol{e}_1, \boldsymbol{e}_2, \boldsymbol{e}_3 \in \mathbb{R}^d$, each selecting a distinct third of the features:

$$\boldsymbol{e}_1 = [\underbrace{1, \ldots, 1}_{d/3}, \underbrace{0, \ldots, 0}_{2d/3}],$$

$$\boldsymbol{e}_2 = [\underbrace{0, \ldots, 0}_{d/3}, \underbrace{1, \ldots, 1}_{d/3}, \underbrace{0, \ldots, 0}_{d/3}],$$

$$\boldsymbol{e}_3 = [\underbrace{0, \ldots, 0}_{2d/3}, \underbrace{1, \ldots, 1}_{d/3}].$$

The graph-level label is then computed as:

$$y = \mathbb{I}\left(\sin(\boldsymbol{x}^\top \boldsymbol{e}_1) \cdot \cos(\boldsymbol{x}^\top \boldsymbol{e}_2) + (\boldsymbol{x}^{\circ 2})^\top \boldsymbol{e}_3 + \varepsilon > 0\right), \quad \varepsilon \sim \mathcal{N}(0, 0.1),$$

where $\boldsymbol{x}^{\circ 2}$ denotes the element-wise square of $\boldsymbol{x}$, i.e., the Hadamard power.

## A9 EXPERIMENTAL SETTING AND EFFICIENCY ANALYSIS

We evaluate the computational complexity and efficiency of MGMT through both theoretical and empirical analysis. This section is structured as follows: Section A9.1 presents a theoretical runtime complexity analysis of MGMT's core components; Section A9.2 provides empirical scalability results across four key input dimensions; Section A9.3 offers runtime profiling and efficiency comparisons, including infrastructure details and training costs.

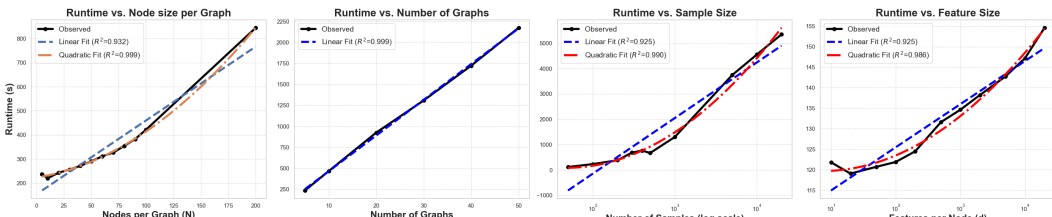

Figure A7: Scalability analysis of MGMT with respect to key input parameters. We evaluate the empirical runtime of MGMT under controlled variations of (i) number of nodes per graph ($N$), (ii) number of graphs per sample ($n$), (iii) number of samples (log scale), and (iv) feature dimensionality ($d$).Runtime scales quadratically with $N$ due to the dense self-attention in the graph-specific Graph Transformers ($\mathcal{O}(N^2 \cdot d)$), and linearly with $n$, confirming the modular and scalable design of MGMT. Sample size and feature dimension contribute to runtime growth in accordance with expectations, with minor deviations at small scales. Linear and quadratic regression fits are shown for interpretability, along with corresponding $R^2$ values.

### A9.1 THEORETICAL COMPLEXITY ANALYSIS.

The total computational complexity of MGMT is governed by three main components: (1) graph-specific Transformer encoders, (2) meta-graph construction, and (3) the final meta-graph Transformer.

**Graph-specific Transformer encoders**   The computational complexity depends on the GT backbone choice. For a graph $\mathcal{G}_i$ with $N_i$ nodes and $d$-dimensional features, the **GAT backbone** in Section 3.1.1, attention is restricted to local neighborhoods, yielding $\mathcal{O}(|\mathcal{E}_i|d)$ per graph $\mathcal{G}_i$ with $|\mathcal{E}_i|$ edges and $d$-dimensional features, totaling $\mathcal{O}(n|\mathcal{E}|d)$ across $n$ graphs. For **dense attention** (e.g., Graphormer Ying et al. (2021)), every node attends to all others, resulting in $\mathcal{O}(N_i^2 d)$ per graph with $N_i$ nodes, or $\mathcal{O}(nN^2 d)$ total. For **sparse attention** (e.g., top-$K$ Zhao et al. (2019)), where each node attends to $K \ll N_i$ neighbors, the complexity is $\mathcal{O}(N_i K d)$ per graph, or $\mathcal{O}(nNKd)$ total.

**Meta-graph construction**   Two steps: (a) supernode extraction by scoring and thresholding nodes is $\mathcal{O}(N_i)$ per graph, totaling $\mathcal{O}(nN)$; (b) superedge creation computes pairwise similarities among selected supernodes. Let $S_i$ be supernodes in graph $i$ and $S_{\text{total}} = \sum_i S_i$. This step costs $\mathcal{O}(S_{\text{total}}^2 d)$, i.e., $\mathcal{O}(n^2 S^2 d)$ for roughly $S$ per graph, with $S_i \ll N_i$.

**Meta-graph Transformer**   Applied over $S_{\text{total}}$ supernodes, yielding $\mathcal{O}(S_{\text{total}}^2 d)$ (approximately $\mathcal{O}(n^2 S^2 d)$).

The dominant term is the per-graph encoder, $\sum_i \mathcal{O}(N_i^2 d)$. Meta-graph construction and inference operate on a much smaller set of supernodes ($S_{\text{total}} \ll \sum_i N_i$) and thus are comparatively lightweight. Quadratic factors at the meta-graph level are in $S_{\text{total}}$ (and $n$), which remains moderate by design.

### A9.2 SCALABILITY ANALYSIS

To validate the theoretical complexity discussed in Section A9.1, we empirically evaluated the runtime behavior of MGMT with respect to four key input parameters: number of nodes per graph ($N$), number of graphs per sample ($n$), number of samples, and node feature dimensionality ($d$). In each experiment, we fixed the model architecture, training epochs (100), and batch size to enable consistent runtime comparisons, and reported runtimes averaged over 10 independent runs. Results in Figure A7 align with theory and show efficient scaling.

**Runtime vs. Nodes per Graph ($N$).**   As predicted by the $\mathcal{O}(N^2 \cdot d)$ complexity of Transformer-based attention, the observed runtime increases superlinearly with $N$. The curve aligns closely with a quadratic fit ($R^2 = 0.999$), reflecting the cost of dense all-pairs attention in graph-specific encoders.

**Runtime vs. Number of Graphs per Sample ($n$).**   The runtime grows approximately linearly with $n$, validating the modular structure of MGMT where graph-specific encoders operate in parallel and the size of the meta-graph remains bounded. This confirms that MGMT scales well with respect to the number of graphs in practical regimes and supports our theoretical analysis in Section A9.1.

**Runtime vs. Number of Samples.** We observe a near-quadratic growth in runtime (on a log scale) as the number of samples increases, consistent with expectations. This is attributed to repeated forward passes and meta-graph construction across samples, particularly in mini-batch training settings.

**Runtime vs. Feature Dimensionality ($d$).** Despite the theoretical linear dependence on $d$ in attention layers, the empirical curve remains nearly flat. This is due to early feature compression in MGMT's architecture, which transforms high-dimensional node features into a lower-dimensional latent space prior to attention and reasoning steps.

## A9.3 RUNTIME PROFILING AND MODEL EFFICIENCY

Building on the complexity analysis and scalability trends in Section A9.2, we profile per-epoch runtime to isolate the cost of each architectural component. Table A4 reports average epoch times for MGMT and graph-attention baselines (those that perform graph reasoning and/or meta-graph fusion).

**Baselines** MGMT's meta-graph reasoning adds minimal overhead: it is faster than MMGL on all datasets except LFP, despite including supernode detection and adaptive depth. Ablations that remove intra-graph edges or the meta-graph yield small speedups but reduce accuracy (see Table A2), illustrating a speed–accuracy trade-off.

MultiMoDN, MedFuse, and FlexCare are omitted from Table A4 because they do not use graph representations or attention; direct runtime comparison to graph-based models would be misleading. These methods operate on tabular inputs with shallow fusion, yielding lower computational cost by design but consistently lower accuracy than MGMT (Table A2).

Table A5 decomposes MGMT's epoch time into data preparation, graph encoders, supernode/superedge construction, meta-graph formation, and the final classifier. The dominant cost is the graph Transformer encoder, consistent with the $\mathcal{O}(N^2 d)$ complexity; meta-graph construction and reasoning are comparatively lightweight due to the compact meta-graph.

Overall, MGMT balances expressivity and efficiency: it achieves higher accuracy than non-graph and shallow fusion baselines while maintaining practical per-epoch runtimes.

**Compute Infrastructure and Training Cost.** All experiments were conducted on a shared CPU-based server provided by our lab. Each training job utilized 4 parallel CPU workers and approximately 4 GB of RAM. No GPU resources were used.

For baseline experiments, we trained a total of 250 models. Each model took on average 5.5 hours to train, amounting to approximately **1,375 CPU hours**.

For MGMT model training and hyperparameter tuning, the total compute time was as follows:

- **LFP dataset:** 100 Optuna trials, each taking 71 minutes on average, resulting in approximately **118.3 CPU hours**
- **Alzheimer dataset:** 100 Optuna trials, each taking 5 hours and 18 minutes on average, resulting in approximately **530 CPU hours**
- **Simulation Setting 1:** 50 iterations, each taking 29 minutes on average, resulting in approximately **24.2 CPU hours**
- **Simulation Setting 2:** 50 iterations, each taking 31 minutes on average, resulting in approximately **25.8 CPU hours**
- **Simulation Setting 3:** 50 iterations, each taking 49 minutes on average, resulting in approximately **40.8 CPU hours**

In total, MGMT-related training required approximately **739 CPU hours**. Additional compute time spent on development, debugging, and model refinement was not recorded.

Table A4: Comparison of average epoch runtime (in seconds) between various meta-graph configurations and baseline models across each dataset.

| Model Variant | Alzheimer | LFP Data | Experiment 1 | Experiment 2 | Experiment 3 |
|---|---|---|---|---|---|
| MMGL | 174.23 | **63.12** | 21.85 | 29.02 | 33.98 |
| MGMT w/o Meta-Graph and Adaptive Depth | 174.10 | 64.33 | **15.10** | **17.20** | **32.60** |
| MGMT w/o Intra-graph Edges | 156.77 | 63.69 | 15.72 | 18.83 | 32.71 |
| MGMT w/o Supernode Selection | 215.46 | 59.61 | 19.91 | 19.31 | 35.61 |
| MGMT | **162.93** | 67.33 | 16.67 | 17.59 | 33.01 |

Table A5: Detailed epoch running time (in seconds) for the MGMT model across different datasets.

| Dataset | Total | Data Prep | Graph-specific encoding | SuperEdge & Node Extraction | Meta-Graph | Final Model |
|---|---|---|---|---|---|---|
| Alzheimer | 162.93 | 1.81 | 119.24 | 28.64 | 1.56 | 13.18 |
| LFP Data | 67.33 | 0.88 | 59.74 | 1.38 | 1.19 | 1.25 |
| Experiment 1 | 16.67 | 0.23 | 16.26 | 0.07 | 0.06 | 0.05 |
| Experiment 2 | 17.59 | 0.44 | 16.40 | 0.26 | 0.25 | 0.24 |
| Experiment 3 | 33.01 | 0.51 | 32.25 | 0.09 | 0.08 | 0.08 |

## A10 SENSITIVITY ANALYSIS OF HYPERPARAMETERS

The MGMT framework includes several hyperparameters that influence model performance and computational efficiency. In this section, we investigate the sensitivity of two key hyperparameters: the attention score threshold ($\tau$) used for supernode selection, and the cosine similarity threshold ($\gamma$) used in inter-graph edge construction.

### A10.1 ATTENTION SCORE THRESHOLD (SUPERNODE SELECTION)

To assess the impact of $\tau$, we conducted a controlled experiment on synthetic data generated under Setting 1 (see Appendix A8). We have a total of 100 samples and 5 graphs per each sample, where each graph consists of 10 nodes, with 30 features per node. We trained all models for 100 epochs and averaged accuracy and runtime over 50 repetitions.

Intuitively, decreasing $\tau$ results in more nodes being selected as supernodes, increasing computational cost and potentially introducing noisy or redundant information. In contrast, higher thresholds select fewer supernodes, reducing runtime but possibly discarding useful information. As shown in Figure A8 left panel, the runtime decreases steadily as $\tau$ increases, which aligns with the reduced number of supernodes and associated computations. However, model accuracy shows a non-monotonic trend: it peaks at $\tau = 0.3$ (64.5%) and declines on either side. This behavior illustrates a tradeoff between overfitting (when too many nodes are included) and information loss (when too few nodes are retained).

### A10.2 COSINE SIMILARITY THRESHOLD (INTER-GRAPH EDGE CONSTRUCTION)

Moreover, to assess the effect of the cosine similarity threshold $\gamma$ used for inter-graph edge construction, we performed a controlled sensitivity analysis using synthetic data generated under Setting 1 (see Appendix A8). We have a total of 100 samples and 5 graphs per each sample, where each graph consists of 100 nodes, with 30 features per node. All models were trained for 100 epochs, and both accuracy and runtime were averaged over 50 repetitions.

As shown in Figure A8 right panel, runtime remains largely stable across different $\gamma$ values, indicating that inter-graph edge density has minimal impact on computational overhead since meta-graph construction occurs post graph-specific encoding and operates over a reduced number of supernodes.

Accuracy, however, demonstrates a non-monotonic trend. When $\gamma$ is very small, the meta-graph becomes fully connected, enabling the model to consider all potential inter-graph interactions. Although this theoretically maximizes expressiveness (since attention-based transformers can learn to prioritize relevant connections), it increases the risk of overfitting due to the inclusion of noisy or spurious edges. On the other hand, when $\gamma$ is close to 1, the meta-graph becomes sparse or even disconnected, leading to an underutilization of cross-graph dependencies.

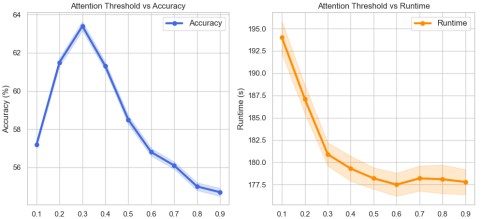 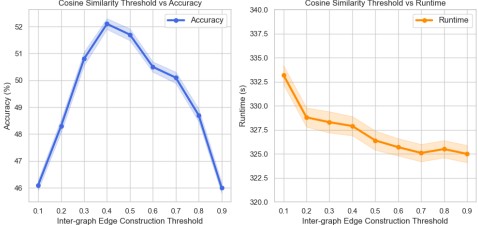

Figure A8: Sensitivity analysis of two key hyperparameters in the MGMT framework. **(a) Left two plots:** The attention score threshold $\tau$ controls supernode selection. Lower thresholds include more nodes, increasing runtime and potentially introducing noise, while higher thresholds risk discarding informative nodes. Accuracy peaks at $\tau = 0.3$, suggesting a balance between expressiveness and overfitting. **(b) Right two plots:** The cosine similarity threshold $\gamma$ governs inter-graph edge construction in the meta-graph. Accuracy peaks at moderate values of $\gamma$, reflecting a trade-off between dense connectivity (risking overfitting) and sparsity (losing cross-graph interactions). Runtime remains largely stable across $\gamma$, as meta-graph construction operates over a small number of supernodes.

The highest accuracy occurs at intermediate values (e.g., $\gamma = 0.4$), suggesting that retaining only the most semantically meaningful inter-graph links allows the model to balance expressiveness with robustness. These findings reinforce the results from our ablation studies (Figure 2), which demonstrate that incorporating carefully selected inter-graph edges substantially improves downstream performance.

## A11 IMPACT OF SIMILARITY METRICS IN META-GRAPH CONSTRUCTION

The construction of inter-graph edges in the meta-graph relies on computing pairwise similarities between node embeddings extracted from different graphs. While cosine similarity is commonly adopted due to its scale-invariant properties, other alternatives such as Pearson correlation, Euclidean distance, and dot product, may also be used to define similarity across nodes. This section evaluates the extent to which the choice of similarity metric affects downstream performance.

To investigate this, we conducted a controlled experiment on a synthetic dataset generated under Setting 1 (see Appendix A8). For each similarity function, we compute full cross-graph similarity matrices between node embeddings and apply a fixed top-$k$ rule with $k = 10$ to select inter-graph edges, ensuring identical sparsity across metrics. Each configuration is run 50 times; we report the mean accuracy.

We compare cosine similarity, Pearson correlation, negative Euclidean distance converted to similarity via $1/(1 + d_{ij})$, and dot product. Results show modest but consistent differences: dot product attains the highest accuracy (0.661), followed by cosine (0.654), Pearson(0.648), and Euclidean (0.642). The spread is small (1.9 percentage points), indicating limited sensitivity to the similarity choice under this setup.

## A12 SINGLE-GRAPH AND MULTI-GRAPH RESULTS WITH ALTERNATIVE GT BACKBONES

In the main text, we implement MGMT with a localized GAT-style GT backbone. To verify that MGMT is not tied to this particular choice and to better understand the role of local versus global attention, we conducted two sets of complementary experiments. First, we replaced GAT with several state-of-the-art GT variants in a *single-graph* setting on the LFP neuroscience dataset, with and without our depth-aware aggregation. Second, we implemented MGMT with different depth-aware GT backbones used both as per-graph feature encoders and as the final feature learning and prediction module on the meta-graph. This section reports and analyzes these results.

Table A6: Comparison of test accuracy between GT variations (GraphGPS, GRIT, Exphormer, EGT, GAT) and their depth-aware counterparts on the LFP dataset (single-graph setting).

| Models | SuperChris | Barat | Stella | Mitt | Buchanan |
|---|---|---|---|---|---|
| GraphGPS | $38.58 \pm 0.09$ | $31.45 \pm 1.39$ | $37.42 \pm 1.01$ | $31.10 \pm 1.66$ | $35.32 \pm 1.81$ |
| Depth-aware GraphGPS aggregation | $39.03 \pm 1.76$ | $30.97 \pm 1.17$ | $36.45 \pm 1.31$ | $30.65 \pm 1.57$ | $37.74 \pm 2.07$ |
| GRIT | $35.76 \pm 1.71$ | $31.77 \pm 1.85$ | $40.61 \pm 0.43$ | $30.58 \pm 1.96$ | $32.64 \pm 2.04$ |
| Depth-aware GRIT aggregation | $38.40 \pm 2.50$ | $31.63 \pm 1.98$ | $40.18 \pm 1.09$ | $32.12 \pm 1.39$ | $34.16 \pm 2.29$ |
| Exphormer | $40.23 \pm 1.45$ | $32.26 \pm 1.77$ | $35.65 \pm 1.67$ | $29.84 \pm 1.58$ | $35.65 \pm 2.08$ |
| Depth-aware Exphormer aggregation | $\mathbf{42.04 \pm 1.65}$ | $38.54 \pm 1.42$ | $39.17 \pm 2.16$ | $34.02 \pm 1.71$ | $39.50 \pm 1.98$ |
| EGT | $40.23 \pm 1.95$ | $33.23 \pm 1.36$ | $36.77 \pm 2.19$ | $28.71 \pm 1.52$ | $38.06 \pm 1.60$ |
| Depth-aware EGT aggregation | $40.06 \pm 1.36$ | $32.74 \pm 1.88$ | $38.87 \pm 1.56$ | $33.71 \pm 2.10$ | $\mathbf{40.97 \pm 1.03}$ |
| GAT | $33.16 \pm 1.08$ | $32.46 \pm 1.34$ | $35.57 \pm 1.42$ | $30.52 \pm 1.42$ | $31.42 \pm 1.96$ |
| Depth-aware GAT aggregation | $36.42 \pm 1.71$ | $\mathbf{40.43 \pm 1.09}$ | $\mathbf{40.31 \pm 1.52}$ | $\mathbf{34.79 \pm 1.41}$ | $34.17 \pm 1.91$ |

### A12.1 SINGLE-GRAPH LFP EXPERIMENTS WITH DIFFERENT GT BACKBONES

Table A6 compares test accuracy on the LFP dataset when training single-graph models separately on each animal using different GT backbones, either in their vanilla form or augmented with our depth-aware aggregation mechanism. Across all GT backbones tested (local, global, and sparse), the depth-aware version consistently improves performance over the corresponding vanilla backbone. The relative gains are largest for backbones whose effective receptive field is more local (GAT) or sparsified (Exphormer, GRIT, EGT), where depth-aware aggregation compensates for limited single-layer reach by mixing information across multiple depths. In contrast, GraphGPS, which already combines local message passing and global attention with strong residual connections, benefits only marginally from our depth-aware aggregation. These results support the claim that depth-aware aggregation is a generic, backbone-agnostic enhancement and not specific to GAT.

Replacing GAT with more advanced GT backbones (GraphGPS, GRIT, Exphormer, EGT) yields modest but consistent gains at the single-graph level, confirming that the LFP task does benefit from long-range or sparse global attention when graphs are treated independently. However, depth-aware aggregation narrows this gap substantially: depth-aware GAT becomes competitive with depth-aware Exphormer and depth-aware EGT, showing that our proposed $L$-hop mixing mechanism is often as important as the specific GT backbone.

### A12.2 MGMT WITH DIFFERENT GT BACKBONES

We next plug these depth-aware GT backbones into MGMT and evaluate in the multi-graph regime. Here, each depth-aware GT backbone has been used both as a per-graph encoder and as the final feature learning and prediction module on the meta-graph. Table A7 reports the results on the LFP, Alzheimer, and simulation datasets.

On the LFP dataset, all depth-aware MGMT variants based on GAT, GraphGPS, GRIT, and Exphormer attain very close accuracies (41.86–42.24%), with absolute differences below 0.4 points. A similar pattern holds for the Alzheimer dataset, where the best-performing variants (depth-aware GAT and depth-aware Exphormer) achieve nearly indistinguishable accuracies within their reported uncertainty. Thus, once we move to the multi-graph regime and apply MGMT's meta-graph fusion, the specific choice of GT backbone becomes substantially less critical than in the single-graph setting.

These experiments clarify the relationship between GT expressivity and MGMT's meta-graph mechanism. In MGMT, the core object is the meta-graph built from supernodes and superedges. Supernodes are defined via depth-aggregated attention on edges: for each node $u$ we use the score

$$\sum_{(u,v) \in E} \alpha_{uv},$$

where the $\alpha_{uv}$ are learned and updated at every layer but are always computed with respect to the underlying edge set $E$. With localized GAT-style attention, $E$ consists only of the true graph edges (plus self-loops), so a high supernode score has a clear meaning: node $u$ sends strong attention along its *real* anatomical or structural connections. This is exactly the semantics MGMT needs when it selects supernodes, defines superedges, and constructs a meta-graph that is intended to reflect task-relevant structure in the original LFP/MRI networks.

Table A7: Comparison of test accuracy between MGMT variants created using different GT variants for feature encoding and Final Feature Learning and Prediction (Depth-aware GAT aggregation, Depth-aware GraphGPS, Depth-aware GRIT, Depth-aware Exphormer, and Depth-aware EGT aggregation)

| Models | LFP Dataset | Alzheimer | Experiment 1 | Experiment 2 | Experiment 3 |
|---|---|---|---|---|---|
| GAT | 40.64 ± 0.223 | 81.20 ± 0.85 | 64.20 ± 2.40 | 68.80 ± 1.17 | 71.45 ± 0.57 |
| Depth-aware GAT aggregation | **42.13 ± 0.252** | 83.11 ± 0.84 | **65.47 ± 2.39** | **69.90 ± 1.19** | **73.21 ± 0.59** |
| Depth-aware GraphGPS aggregation | 41.94 ± 0.162 | 80.12 ± 0.67 | 62.37 ± 1.56 | 67.49 ± 0.98 | 72.38 ± 0.83 |
| Depth-aware GRIT aggregation | 42.24 ± 0.345 | 82.59 ± 1.03 | 61.06 ± 2.31 | 69.23 ± 1.01 | 72.98 ± 0.61 |
| Depth-aware Exphormer aggregation | 41.86 ± 0.427 | **83.29 ± 0.58** | 62.93 ± 1.92 | 67.46 ± 1.32 | 72.46 ± 0.49 |
| Depth-aware EGT aggregation | 40.08 ± 0.162 | 81.79 ± 1.26 | 60.52 ± 1.86 | 68.96 ± 0.94 | 71.30 ± 0.49 |

Several SOTA GT backbones, however, substantially modify this edge set. EGT and Graphormer-style models effectively allow attention between all node pairs, and Exphormer adds expander edges and virtual node connections. In these cases, $E$ contains a mix of true and artificial edges, so $\sum_{(u,v)\in E} \alpha_{uv}$ blends attention along physical connections and model-constructed links. This is often beneficial for single-graph prediction (hence the stronger vanilla GT backbones in Table A6), but it dilutes the structural meaning of supernodes and superedges and can "contaminate" the meta-graph by injecting artificial connectivities, which is undesirable in MGMT's interpretability-driven setting. GRIT, on the other hand, provides an instructive intermediate case. Its design combines global, kernelized attention with a sparsified connectivity pattern that is optimized for single-graph prediction, and in the vanilla setting, this yields clear gains over GAT (Table A6). In MGMT, however, GRIT's sparsified but topology-modified attention pattern slightly alters the edge set used for supernode scoring and superedge construction, so the resulting meta-graph is not systematically better aligned with the underlying anatomical or structural connectivity than the one induced by localized GAT attention. As a result, GRIT achieves similar but not consistently superior performance to depth-aware GAT within the MGMT framework, despite its advantage in the single-graph regime.

We therefore observe a trade-off. Global/sparse GT backbones can be slightly stronger in single-graph tasks, whereas topology-preserving localized attention is better aligned with MGMT's goal of building an interpretable meta-graph from true edges. Empirically, once depth-aware aggregation and meta-graph fusion are enabled, MGMT with depth-aware GAT, GraphGPS, GRIT, and Exphormer all achieve very similar performance (Table A7), and accuracy is stable across a range of supernode thresholds $\tau$ (Appendix A10). This indicates that the main gains in MGMT come from depth-aware multi-scale mixing, and more importantly the supernode/meta-graph construction that explicitly encodes and interprets cross-graph connections, rather than from a particular GT variant.

Overall, Tables A6 and A7 highlight that (i) MGMT is backbone-agnostic; stronger GT backbones do yield better single-graph performance, and (ii) With the help of meta-graph construction, MGMT is relatively robust to the choice of backbone in the multi-graph fusion setting.

In terms of efficiency, we also measured average epoch runtime (in seconds) for MGMT implemented with each backbone on the Alzheimer dataset (Table A8). Depth-aware GAT remains among the most efficient MGMT variants overall. GRIT attains the lowest total per-epoch time (158.2s vs. 162.9s for GAT) by substantially reducing the cost of the graph-specific encoder (102.8s vs. 119.2s), but this saving is largely offset by a more expensive SuperEdge & Node Extraction stage (40.4s vs. 28.6s). This increase is consistent with GRIT's design: its edge-aware, relation-augmented attention produces denser and more heterogeneous attention patterns than localized GAT, so MGMT must process more non-negligible attention coefficients when aggregating edge weights, selecting supernodes, and constructing superedges across layers. GraphGPS, Exphormer, and EGT all lead to higher average per-epoch runtimes than GAT (171.5–187.3s per epoch), because their more global or hybrid attention mechanisms generate richer attention maps that MGMT must export, aggregate, and threshold at every layer, increasing the cost of both the encoder and the SuperEdge & Node Extraction block (41.9–46.0s vs. 28.6s for GAT). Overall, these results indicate that while advanced GT backbones modestly change the balance between encoder and meta-graph construction costs, they do not usually yield faster end-to-end MGMT training than the depth-aware GAT variant. For these reasons, we present MGMT with a GAT backbone in the main paper as a balanced choice in terms of simplicity, interpretability, runtime, and performance.

Table A8: Per-epoch runtime breakdown (in seconds) for MGMT with different depth-aware GT backbones on the Alzheimer dataset.

| Backbone (MGMT variant) | Total | Data Prep | Graph-specific encoding | SuperEdge & Node Extraction | Meta-Graph | Final Model |
|---|---|---|---|---|---|---|
| Depth-aware GAT aggregation | 162.93 | 1.81 | 119.24 | 28.64 | 1.56 | 11.68 |
| Depth-aware GraphGPS aggregation | 171.50 | 1.85 | 114.10 | 41.85 | 1.64 | 12.06 |
| Depth-aware GRIT aggregation | 158.20 | 1.82 | 102.75 | 40.43 | 1.60 | 11.60 |
| Depth-aware Exphormer aggregation | 180.70 | 1.90 | 118.75 | 45.95 | 1.70 | 12.40 |
| Depth-aware EGT aggregation | 187.30 | 1.88 | 126.30 | 44.55 | 1.68 | 12.89 |

## A13 DYNAMIC, DISTRIBUTION-BASED THRESHOLDING FOR SUPERNODES AND META-GRAPH EDGES

In the main text, MGMT uses two scalar thresholds: (i) $\tau$ for supernode extraction based on attention scores, and (ii) $\gamma$ for sparsifying inter-graph edges in the meta-graph using cosine similarity. These thresholds are tuned on validation data together with standard hyperparameters (learning rate, depth, dropout, etc.). Section 3.1.2 and Appendix A10 already show that MGMT is stable across a wide range of $\tau$ and $\gamma$ values.

Here, we additionally implemented and evaluated a *dynamic, distribution-based thresholding* variant of MGMT, in which the effective thresholds are determined from the empirical distributions of scores rather than being validation-tuned scalars.

**Dynamic supernode selection (data-driven $\tau$).** Let $a_v$ denote the aggregated attention score for node $v$ in a given graph (obtained from the TransformerConv layers as in Section 3.1.1). We first normalize the node-level attention scores *within each graph*:

$$\tilde{a}_v = \frac{a_v - \min_u a_u}{\max_u a_u - \min_u a_u + 10^{-5}} \in [0, 1].$$

Instead of specifying a validation-tuned $\tau$, we choose a retention rate $\rho_{\text{sup}} \in (0, 1)$ (e.g., $\rho_{\text{sup}} = 0.3$) and keep only the top $\rho_{\text{sup}}$ fraction of nodes according to $\tilde{a}_v$. Concretely, for each graph $g$ we compute the $(1 - \rho_{\text{sup}})$-quantile $c_{\text{sup},g}$ of its normalized scores and define the supernode set as

$$\mathcal{S}_g = \{v : \tilde{a}_v \geq c_{\text{sup},g}\}.$$

Equivalently, this induces a *graph-specific, data-driven threshold*

$$\tau_g = c_{\text{sup},g},$$

which depends on the empirical distribution of attention scores in graph $g$. If the attention scores in a new dataset are more concentrated or more diffuse, the resulting $\tau_g$ automatically adjusts.

**Dynamic meta-graph edge construction (data-driven $\gamma$).** Given the set of supernodes across all graphs, we compute the cosine similarity matrix over their embeddings:

$$e_{uv} = \frac{\boldsymbol{H}_u^\top \boldsymbol{H}_v}{\|\boldsymbol{H}_u\|\|\boldsymbol{H}_v\|} \tag{A28}$$

In the original MGMT formulation, inter-graph edges are obtained by applying a validation-tuned parameter $\gamma$. In the dynamic variant, we specify a retention rate $\kappa_{\text{edge}} \in (0, 1)$ (e.g., $\kappa_{\text{edge}} = 0.05$) and keep the top $\kappa_{\text{edge}}$ fraction of all off-diagonal similarities by defining threshold as the $(1 - \kappa_{\text{edge}})$-quantile of the entries of the cosine similarity matrix. Now $\gamma$ is *not validation-tuned*: it is induced by the empirical distribution of similarities in the current dataset.

**Experimental comparison and discussion.** Table A9 compares the original validation-tuned threshold MGMT (with $\tau$ and $\gamma$ selected via validation) to the dynamic quantile-based variant described above. We observe that the dynamic variant attains performance that is close to the best validation-tuned threshold configuration across all datasets, with a small decrease in accuracy in some cases.

This behavior is natural: Previously $\tau, \gamma$ were tuned specifically to maximize validation performance on each dataset, while the dynamic variant applies a generic, model-agnostic rule that does not

Table A9: Comparison of MGMT with validation-tuned thresholds vs. dynamic, distribution-based thresholds for supernode selection and meta-graph edge construction. Values are mean $\pm$ standard deviation over 5 folds.

| Dataset | MGMT (validation-tuned $\tau, \gamma$) | MGMT (dynamic, quantile-based) |
|---|---|---|
| LFP dataset | $42.13 \pm 0.25$ | $41.26 \pm 0.73$ |
| Alzheimer | $83.11 \pm 0.84$ | $81.96 \pm 0.76$ |
| Experiment 1 | $65.47 \pm 2.39$ | $63.35 \pm 1.64$ |
| Experiment 2 | $69.90 \pm 1.19$ | $67.72 \pm 1.24$ |
| Experiment 3 | $73.21 \pm 0.59$ | $71.96 \pm 1.23$ |

exploit dataset-specific optimal sparsity levels. Consequently, a minor loss in accuracy is a reasonable trade-off for eliminating validation-tuned thresholds and making sparsification fully data-driven. Importantly, both the sensitivity analysis in Appendix A10 and the results in Table A9 indicate that MGMT does *not* hinge on finely tuned thresholds: it remains robust under both validation-tuned and distribution-based thresholding schemes.

## A14  CAUSAL EXTENSIONS OF MGMT VIA CAL

Recent advances in causal graph learning offer promising directions for identifying causally important nodes. Causal masking approaches such as CAL (Sui et al., 2022) learn to disentangle causal from spurious features through an intervention-based training scheme. Counterfactual methods reviewed in Guo et al. (2025) identify critical graph components by measuring prediction changes under minimal graph edits. Next we present one concrete potential design, while future work can explore additional promising directions for causal learning in multi-graph settings.

**CAL (Causal Attention Learning)** Sui et al. (2022) offers a natural integration with our framework. CAL introduces a disentanglement module to separate causal features that reflect intrinsic graph properties from shortcut features arising from data biases or trivial patterns. Specifically, given a graph $\mathcal{G} = (\mathcal{V}, \mathcal{E})$ with $N = |\mathcal{V}|$ nodes, feature matrix $\boldsymbol{X} \in \mathbb{R}^{N \times d}$, and adjacency matrix $\boldsymbol{A} \in \mathbb{R}^{N \times N}$, CAL learns two masks $\boldsymbol{M}^a \in \mathbb{R}^{N \times N}$ (for edges) and $\boldsymbol{M}^x \in \mathbb{R}^{N \times 1}$ (for node features) via causal intervention (see Sui et al. (2022) for details). Each element of the masks, with value in $(0, 1)$, indicates the causal relevance to the label. Applying CAL to each graph $i \in [n]$ yields causal masks $\boldsymbol{M}_i^a$ and $\boldsymbol{M}_i^x$. We can re-weight the edge attention scores in (3), section 3.1.1 by $\boldsymbol{M}_i^a$:

$$\boldsymbol{\alpha}_i^{\text{caus}} = \boldsymbol{\alpha}_i \odot \boldsymbol{M}_i^a,$$

where $\odot$ denotes the Hadamard product. Similarly, we can adjust the supernode selection rule in (4) by

$$\mathcal{S}_i = \left\{ u \in \mathcal{V}_i \mid M_{i,u}^x \sum_{(u,v) \in \mathcal{E}_i} \alpha_{i,uv}^{\text{caus}} \geq \tau \right\}.$$

where $M_{i,u}^x$ denotes the causal score for node $u$. This ensures selected supernodes maintain strong causal relationships with the label.

## A15  SPARSE TOP-$k$ ATTENTION MECHANISM APPLIED TO THE GRAPH-SPECIFIC ENCODERS

As discussed in Appendix A9.1, the dominant computational cost in MGMT comes from the graph-specific encoder, whose complexity scales with the number of attended neighbors per node. To further investigate the potential of sparse attention in our framework, we perform a preliminary experiment where we replace the localized GAT-style attention in the graph-specific encoder with *top-k attention*, while leaving the rest of MGMT unchanged.

Concretely, at each GAT layer and for each node $u$ in graph $\mathcal{G}_i$, we compute the standard attention scores $\{\alpha_{uv}\}_{v \in \mathcal{N}(u)}$ over its 1-hop neighbors. We then keep only the $k$ largest-magnitude scores and set the remaining attention weights to zero before normalization. This yields a sparse attention pattern with effective cost $\mathcal{O}(N_i k d)$ per layer (for $N_i$ nodes and $d$-dimensional features), instead of

$\mathcal{O}(|\mathcal{E}_i|d)$. Importantly, we apply this sparsification *only* in the graph-specific encoder; the meta-graph construction and final predictor remain unchanged, but their runtimes are indirectly affected through the change in attention patterns and degrees used for supernode/superedge extraction.

For each configuration, we re-run MGMT end-to-end with the same hyperparameters as in the dense setting and measure per-epoch runtime as well as test accuracy. Figures A9–A10 summarize the total time and accuracy trends as a function of $k$. In each panel, the dashed horizontal line denotes the dense GAT-based MGMT baseline.

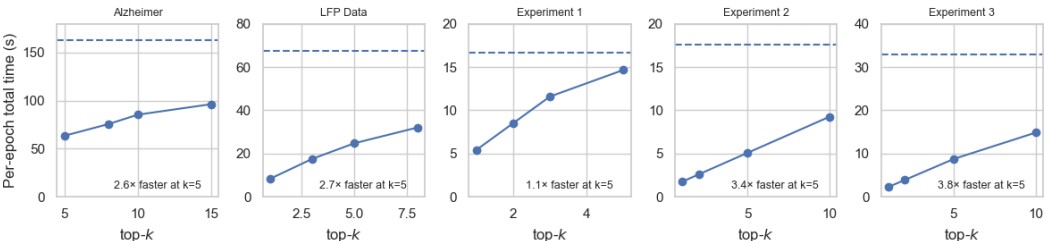

Figure A9: **Effect of top-$k$ attention on MGMT runtime.** Per-epoch total time versus $k$ for each dataset. The dashed line indicates the dense GAT-based MGMT runtime; annotations report the speedup at $k = 5$ relative to this baseline.

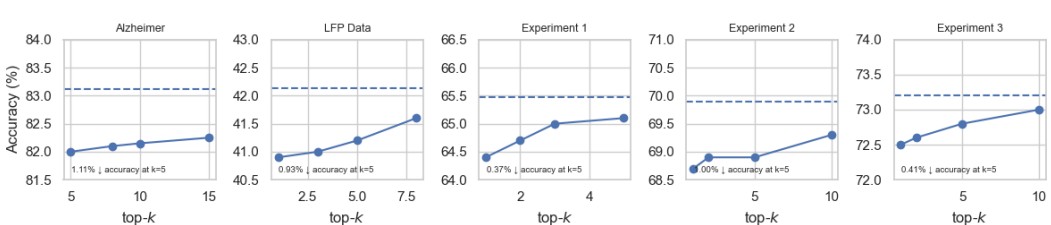

Figure A10: **Effect of top-$k$ attention on MGMT accuracy.** Test accuracy versus $k$ for each dataset. The dashed line indicates the dense MGMT accuracy; annotations report the accuracy change at $k = 5$.

Two patterns emerge from these preliminary results. First, for the Alzheimer and LFP datasets, top-5 attention reduces the per-epoch runtime approximately 2.6 and 2.7 times, respectively, relative to dense MGMT. On the synthetic datasets, speedups at $k = 5$ range from about 1.1 (Experiment 1, with only 5 nodes) to 3.8 (Experiment 3, with 50 nodes) times relative to dense MGMT. As $k$ increases, each node attends to more neighbors, so the encoder cost grows roughly linearly in $k$ (from $\mathcal{O}(Nkd)$ toward the dense limit), and the total runtime curves in Figure A9 smoothly approach the dense baseline.

Second, as shown in Figure A10, these runtime gains come with mild accuracy changes. Very small $k$ values can drop some informative neighbors and under-connect the graphs, leading to an accuracy loss; increasing $k$ restores more of the original neighborhood structure, allowing the encoder to capture richer local context and thus produces a gentle increase in accuracy.

Overall, these experiments support the feasibility of integrating sparse top-$k$ attention into MGMT: by sparsifying only the graph-specific encoder, we obtain speedups on larger graphs while losing some predictive performance. This provides a concrete path toward scaling MGMT to settings with many modalities and/or larger per-graph node sets, complementing the theoretical complexity discussion in Appendix A9.1.

