# OpenReview forum: "Multi-Graph Meta-Transformer (MGMT)"
_ICLR.cc/2026/Conference — Submitted to ICLR 2026_

### Official Review · Reviewer_dTh6 · 2025-10-15

**Soundness:** 3
**Presentation:** 3
**Contribution:** 2
**Rating:** 4
**Confidence:** 4

**Summary:**

This work introduces at the same time the task of multi-graph learning and a hierarchical graph model which incorporates 1-hop attention and multi-graph fusion to tackle this task.

The paper is well-written and easy to follow, and interesting mathematical motivations are provided for some design choices, especially the meta-graph construction.

The contributions appear however limited, and the lack of many influential graph transformer (GT) baselines weakens the experimental section.

**Strengths:**

The paper is well written and easy to read, and the task it proposed to address is original.
I found the theoretical foundation for meta-graph construction (A5) particularly informative.
The experiment on neural activity in rats' brain is original and constitutes a nice opening to more diverse and practical applications.

**Weaknesses:**

The main weakness I see is the superficial consideration of previous graph learning, and especially graph transformer literature.

This is visible in several parts of the manuscript, as detailed below.

The related works section is very short, and several influencial papers are missing, especially regarding the GT literature (see the references [1-5] below), and hierarchical graph models (e.g. [6]). This omission is felt in the method and experiments sections, and overall weakens this work's contribution.

The method section only considers graph attention through the prism of GAT, which is essentially a MPNN layer that uses attention to aggregate neighbors' features. This formulation has been shown to have several limitations, especially regarding its expressivity (it is no more expressive than 1-WL). Global attention, as used in the SOTA GT models listed below, enhances this expressivity beyond 1-WL. From that regard, the local attention mechanism used in this paper appears very limited, especially as memory-efficient GT models exist (see, e.g., [5]).
Several experimental findings hint at the limited expressivity of the 1-hop neighborhood attention:
- The initial increase in accuracy from small $\tau$ values in Figure A8 is likely caused by a larger attention receptive field, i.e. a more expressive attention layer;
- The decreased ablation performances when supernodes are removed highlight the gains from performing attention over larger neighborhoods.
Supernodes partly alleviate this issue, but to the cost of erasing structural information.

Graph model baselines are missing from the experiments. The SOTA multimodal learning baselines are not graph architectures, except one. However the datasets considered here, both real-world and synthetic, are intrinsically graph learning problems. This is highlighted for LFP which contains nodes that "vary in number and identity across subjects", calling for methods that are invariant to node permutations and encode graph structures. The graph nature of the Alzheimer's database is also highlighted by the drop in ablation performances when edges are discarded. In summary, MGMT should be compared with SOTA GT models, instead of generic multimodal fusion models. Or, alternatively, experiments should be conducted on the same dataset as those latter methods (e.g. MIMIC, which is used in both MedFuse and FlexCare).
The benefits of the proposed method could therefore be properly assessed.


[1] Do transformers really perform badly for graph representation?, Ying et al., NeurIPS 21.

[2] Global self-attention as a replacement for graph convolution, Hussain et al., KDD 22.

[3] A generalization of vit/mlp-mixer to graphs, He et al., ICML 23.

[4] Exphormer: Sparse transformers for graphs, Shirzad et al., ICML 23.

[5] Graph inductive biases in transformers without message passing, Ma et al., ICML 23.

[6] Enhancing Graph Transformers with Hierarchical Distance Structural Encoding, Luo et al., NeurIPS 24.

**Questions:**

Some additional questions:

1. You say in RW that MGMT is fundamentally different from multimodal graph learning, but when different graphs in a sample have different features dimension $d$, you will need separate encoders for each modality. How is it then different from multimodal graph learning?

2. How does Theorem 3.3 / L-hop mixing translate in terms of WL expressivity?

3. It seems from Equation 5 that metagraphs (graphs with supernodes) can become disconnected if $\tau$ is too high. Was it problematic in your experiments?

4. I don't understand the aggregation over the batch (sum over $k$) in Equation (A17), even when assuming all samples (graph collections) have the same number of graphs $n$. If the goal is to evaluate the relevance of depth $l$, why not aggregating over all $n$ graphs and $K$ collections?
The dependency on $i$ of $\Gamma_i^l$ in (A18) is unclear either.
It seems in Equation 3 that this dependency is absent, but the dependency on $k$ also seems to be omitted in that section. Can you clarify this?

---

> ### Author Response · Authors · 2025-11-20
> **Rebuttal**
>
> We would like to thank the Reviewer for careful evaluation of our work and for the valuable suggestions for improvements. In the following, we provide responses and clarifications to the weaknesses or questions raised by the Reviewer.
> > **W1:** The main weakness I see is the superficial consideration of previous graph learning...
>
> We appreciate the reviewer's comment. We have substantially expanded our discussion of graph learning and, in particular, graph transformer literature in the revised manuscript. The Related Work section is now a standalone section with a dedicated subsection on Graph Representation Learning, where we explicitly discuss recent GT architectures, including global-attention models, sparse/structure-aware GTs, and hierarchical GT variants (such as the works cited by the reviewer). We clarify how these models extend beyond 1-WL, how they introduce global or sparse attention patterns, and how MGMT can be implemented with these backbones while focusing on the multi-graph fusion problem rather than on proposing yet another single-graph GT layer. Finally, to ensure this richer literature discussion is reflected in the technical parts of the paper, we added new experiments where we replace GAT with several modern GT backbones both at the single-graph level (Table A6) and inside MGMT (Table A8), reported per-epoch runtime breakdown for MGMT with different depth-aware GT backbones on the Alzheimer dataset (Table A8), and clarified why we adopt a localized GT backbone in our main instantiation and how this choice interacts with meta-graph construction and interpretability (Appendix A12). Taken together, these additional experiments clearly situate MGMT within the current landscape of graph learning.
> >**W2:** The method section only considers graph attention through the prism of GAT...
>
> We agree that a single GAT layer is no more expressive than 1-WL. As shown in Appendix A12, replacing GAT with stronger GT variants (GraphGPS, GRIT, Exphormer, EGT) yields modest single-graph gains, but depth-aware aggregation largely closes this gap so that depth-aware GAT is competitive with these backbones (Table A6). In the multi-graph setting, MGMT variants attain very similar accuracies (Table A7), indicating that most improvements come from the supernode/meta-graph mechanism rather than the particular attention backbone. Moreover, GAT-style attention selects supernodes and superedges using true edges, preserving interpretability, whereas global GT backbones mix physical and artificial links and tend to blur the structural meaning of supernodes; and although GRIT can reduce graph-encoding time, exporting and aggregating attention from all layers makes runtimes across MGMT variants comparable (Table A8).
> >**W3:** Graph model baselines are missing from experiments..
>
> We agree that it is important to include more graph-native baselines, and have revised the experiments accordingly. We have now added single-graph GT baselines by replacing GAT with several SOTA GT backbones (Appendix A12), and incorporated new multi-graph baselines tailored to multiple graphs per entity (AMIGO, MaxCorrMGNN, MGLAM) across all datasets; their performance is reported in Table A2 and analyzed in the main results section, also shown below:
>
> **Table A2.** Accuracy (mean ± standard error) for new SOTA GT-based multi-graph baselines across datasets.
> The proposed model is shown in **bold**.
>
> | Category                      | Model         | Alzheimer        | LFP Data          | Experiment 1       | Experiment 2       | Experiment 3       |
> |------------------------------|---------------|------------------|-------------------|--------------------|--------------------|--------------------|
> | GT multi-graph       | AMIGO         | 79.23 ± 1.04     | 38.92 ± 1.32      | 58.68 ± 2.12       | 65.73 ± 0.89       | 70.42 ± 0.79       |
> | GT multi-graph       | MaxCorrMGNN   | 77.43 ± 1.35     | 35.97 ± 2.73      | 56.23 ± 1.32       | 63.72 ± 1.64       | 71.46 ± 1.01       |
> | GT multi-graph       | MGLAM         | 81.29 ± 0.96     | 38.93 ± 1.02      | 61.96 ± 1.04       | 62.29 ± 1.29       | 69.52 ± 0.39       |
> | Proposed model               | **MGMT**      | **83.11 ± 0.84** | **42.13 ± 2.52**  | **65.47 ± 2.39**   | **69.90 ± 1.19**   | **73.21 ± 0.59**   |
>
> These additions ensure that MGMT is directly compared to SOTA graph and multi-graph architectures, not only to generic multimodal fusion models. Across all settings, MGMT outperforms these graph-based baselines, supporting the claim that its meta-graph–based, subgraph-level fusion provides benefits beyond what existing GT and multi-graph models achieve on these neuroscience and Alzheimer tasks. We retain the multimodal baselines (e.g., MedFuse, FlexCare) to show that even when competing against strong vector-level fusion schemes commonly used in clinical applications, graph-native MGMT remains competitive without changing the underlying datasets.

---

> > ### Author Response · Authors · 2025-11-23
> > **Rebuttal, Continued**
> >
> > ***Quesntions***
> > >**Q1:** You say in RW that MGMT is fundamentally different from multimodal graph learning, but..
> >
> > We apologize for the confusion. In MGMT, we always use separate encoders for each graph, regardless of whether their feature dimensions $d$ match or not, so the need for modality-specific encoders is not what distinguishes our setting from multimodal graph learning. The key difference is what happens after encoding. In multimodal graph learning, the standard practice is either to collapse everything into a single unified heterogeneous graph with one global node set or to pool each modality-specific graph into a single embedding vector and then fuse these vectors (e.g., via shared context tokens or attention over modality embeddings). In contrast, MGMT keeps the graphs structurally separate, performs depth-aware encoding within each graph, and then fuses them by constructing an explicit meta-graph over attention-selected supernodes drawn from all graphs of an entity. This meta-graph preserves intra-graph connectivity and introduces interpretable superedges between aligned substructures across graphs, enabling fine-grained, node-/subgraph-level cross-graph message passing rather than only vector-level fusion. Thus, using separate encoders is shared with multimodal graph learning; what fundamentally differentiates MGMT is its structure-preserving, meta-graph–based fusion mechanism on heterogeneous, unaligned graphs.
> > >**Q2:** How does Theorem 3.3 / L-hop mixing translate in terms of WL expressivity?
> >
> > Thank you for this insightful comment, which allows us to clarify the meaning of L-hop mixing and its relation to WL expressivity. In short, $L$-hop mixing does not directly translate into WL expressivity; they are different ways of characterizing the power of graph models.
> >
> > WL expressivity measures *distinguishing power* between non-isomorphic graphs, while $L$-hop mixing measures *approximation quality* for target functions. Our new theoretical analysis shows that MGMT's WL expressivity depends on the GT backbone choice: with GAT backbone, MGMT is upper-bounded by 1-WL (detailed in the ''MGMT with GAT is 1-WL bound'' paragraph in A4.3). Moreover, aggregating multiple depth information does not overcome 1-WL's fundamental limitations since the model still operates on local neighborhood structure. However, this does not diminish the value of depth-aware aggregation. As stated in Definition 3.1, models capable of representing $L$-hop mixing can exactly recover a specific class of target functions that depend on mixed-depth information. This explains the empirical success of multi-depth aggregation as shown in Table A6 in A12.1: $L$-hop mixing provides superior approximation quality regardless of the GT backbone. Furthermore, MGMT's flexible modular framework allows extending beyond 1-WL by replacing the backbone with more expressive variants (discussed in the last paragraph ``Going beyond 1-WL" in A4.3). We have added Appendix A4.3 in the revised manuscript specifically to provide this analysis and to directly address the reviewer’s question about WL expressivity.
> > >**Q3:** It seems from Equation 5 that metagraphs can become disconnected if $\tau$ is too high.
> >
> > We thank the reviewer for this question. In all reported experiments, the threshold $\tau$ is treated as hyperparameters and selected via cross-validation on the training data, so values that would over-prune and disconnect the meta-graph are automatically avoided; we did not observe unstable behavior or performance collapse due to disconnected meta-graphs in practice. In the revision (Appendix A13), we additionally include a new appendix section where we replace these fixed thresholds with data-driven, dynamic thresholds derived from the attention/similarity distributions, and obtain comparable performance, further indicating that MGMT is robust to the choice of $\tau$.
> > >**Q4:**  I don't understand the aggregation over the batch (sum over k) in Equation (A17)...
> >
> > Thank you so much for this comment. The confidence scores in Equation (3) should indeed depend on $i$, the original expression, which omits this dependency was a typo. We have modified the paper accordingly. Furthermore, we clarify two key points below:
> >
> > First, the confidence scores should be graph-specific. Fixing layer $\ell$, we do not aggregate over all $n$ graphs and $k$ collections because each graph can have different informative depth structures due to graph heterogeneity. This heterogeneity motivates computing different confidence scores $\Gamma_i^{(\ell)}$ for each graph $i$, allowing the model to adaptively weight depth information per graph.
> >
> > Second, there is no explicit dependency on individual sample $k$ because $\Gamma_i^{(\ell)}$ aggregates over all samples $k = 1, \dots, K$ in Equations (A16)-(A17). The summation over $k$ evaluates classification error across the entire batch for graph $i$, producing a single scalar $\Gamma_i^{(\ell)}$ per graph and depth.

---

### Official Review · Reviewer_hJwr · 2025-10-27

**Soundness:** 2
**Presentation:** 2
**Contribution:** 1
**Rating:** 4
**Confidence:** 4

**Summary:**

This paper introduces a framework called Multi-Graph Meta-Transformer (MGMT), designed to effectively integrate information from heterogeneous graphs. The method first independently encodes each graph using specific depth-aware Graph Transformer layers. Subsequently, a "meta-graph" is constructed to capture inter-modal relationships. The meta-graph consists of super-nodes from each graph, extracted as nodes with high attention scores. The edges of the meta-graph comprise two parts: intra-modal edges and inter-modal edges, with the latter established based on similarity between super-nodes. Finally, an additional Graph Transformer is applied to this meta-graph to perform downstream tasks.

**Strengths:**

1. It’s a good attempt to build the "meta-graph" to explicitly model inter-graph relationships between heterogeneous graphs.

2. It’s meaningful and interesting to apply MGMT to neuroscience applications.

**Weaknesses:**

1. The motivation of the paper is not clear. Although the authors argue that multi-graph fusion is challenging, the paper does not clearly specify what exact limitations of existing approaches must be solved. It’s still confusing after reading the introduction. Is the main limitation the current multi-graph fusion methods cannot preserve both inter- and intra-graph structural information? In addition, in introduction part, the authors should summarize the contributions of the paper, which highlights the unique technical novelty and the significance.

2. The works mentioned in related works seem not highly related to this paper. Since the focus of the work is on learning across multi-graph (multiple heterogeneous graphs), the paper should compare against prior research specifically in multi-graph learning rather than primarily discuss multimodal graph learning or general multimodal fusion, which involve fundamentally different settings and assumptions. In addition, the paper structure feels somewhat unconventional, it would be better to have a separate section for related work.

3. The authors claim that the graph-specific Transformer encoders produce depth-aware intra-graph representations. However, this appears to be essentially a form of multi-layer attention aggregation, which is conceptually similar to existing multi-layer aggregation techniques such as LayerMix. It remains unclear what exactly “depth-aware” means beyond weighted averaging across Transformer layers. There is currently no experimental or analytical evidence demonstrating that different layers contribute distinct or complementary information. For example, the authors didn't provide any visualization of the confidence scores $Γ^{(ℓ)}$, or an analysis showing different layers contribute meaningfully. Without such results, it is difficult to conclude that the model truly learns depth-specific structural semantics rather than merely fusing multi-level embeddings. Although the ablation experiment removing adaptive depth yields a performance degradation, this alone is insufficient to validate the claimed depth-aware representation learning.

**Questions:**

1. Why related work section does not mention about multi-graph learning, but instead of multimodal graph learning and multimodal fusion?

2. How does the depth-aware GT layers differ from existing multi-layer aggregation techniques?

3. In section 4.2, What do you mean multimodal classification? Also, in Figure 3, what’s the meaning of Modality 1, 2,…?

---

> ### Author Response · Authors · 2025-11-20
> **Rebuttal**
>
> We would like to thank the Reviewer for the careful evaluation of our work and for the valuable suggestions for improvements. In the following, we provide responses and clarifications to the weaknesses or questions raised by the Reviewer.
> >**W1:** The motivation of the paper is not clear...
>
> We appreciate the reviewer’s feedback on the motivation and have clarified both the problem setting and our contributions in the revised manuscript. We now state explicitly in the Introduction that the core limitation of existing approaches is their inability to jointly preserve both intra-graph structure (within each modality/view/subject) and inter-graph structure (how informative subgraphs across graphs interact), especially when each entity is represented by a collection of heterogeneous graphs with unaligned node sets. Most current multimodal and multi-graph methods fuse information only at the level of pooled graph embeddings or shared context tokens, which discards fine-grained topology at fusion time and makes it difficult to see which parts of which graphs actually interact. In contrast, MGMT is explicitly designed for this heterogeneous multi-graph regime: its key innovation is to perform fusion at the subgraph level via an explicitly constructed meta-graph of attention-selected supernodes that preserves within-graph topology, introduces interpretable superedges that connect functionally aligned substructures across graphs, and supports explicit cross-graph message passing on these supernodes. The main limitation we target is precisely the lack of a structure-preserving, interpretable mechanism that captures both intra- and inter-graph dependencies in multi-graph fusion.
> >**W2, Q1:** The works mentioned in related works seem not highly related to this paper. Why related work section does not mention about multi-graph learning..?
>
> We appreciate this comment and have substantially updated both the related work and experimental baselines to better reflect recent advances in multi-graph learning. In the revised Related Work, we now explicitly cover: (1) modern graph transformer backbones (GraphGPS, GRIT, Exphormer, EGT, and HDSE), as state-of-the-art single-graph GTs, to make clear that MGMT is backbone-agnostic and to motivate the additional backbone experiments in Appendix A12; (2) entity-level multi-graph models (AMIGO, MaxCorrMGNN, and MGLAM), which take multiple graphs per entity and are therefore the most comparable to our setting; these are now included as strong baselines in Results section; (3) multi-view GTs with aligned nodes (MGT and MVGT/MVGTrans), cited to clarify how their aligned multi-view assumption differs from MGMT’s heterogeneous, unaligned multi-graph regime; and (4) population-level multi-graph and graph-of-graphs approaches (AMGL, GraphFM, GraphAlign, and SamGoG), used to position MGMT as a complementary, entity-level meta-graph framework. On the experimental side, we have substantially expanded the evaluation to include more recent multi-graph baselines. We incorporated AMIGO, MaxCorrMGNN, and MGLAM as strong multi-graph baselines, with results summarized in Table A2 (and visualized in Figs. 3–4) and referenced in the main Results section, also shown below:
>
> **Table A2.** Accuracy (mean ± standard error) for different models across datasets, grouped by model family. Newly introduced SOTA GT-based multi-graph baselines are shown in ***bold italics***. The proposed model is shown in **bold**.
>
> | Category                    | Model         | Alzheimer        | LFP Data          | Experiment 1       | Experiment 2       | Experiment 3       |
> |----------------------------|---------------|------------------|-------------------|--------------------|--------------------|--------------------|
> | General-purpose multimodal | MMGL          | 79.38 ± 0.52     | 39.28 ± 1.93      | 59.20 ± 1.04       | 62.80 ± 0.84       | 68.75 ± 0.12       |
> | General-purpose multimodal | MultiMoDN     | 76.44 ± 0.75     | 37.82 ± 1.82      | 60.40 ± 1.67       | 61.50 ± 1.01       | 65.10 ± 0.50       |
> | General-purpose multimodal | FlexCare      | 76.14 ± 0.79     | 36.42 ± 1.88      | 61.10 ± 1.39       | 69.82 ± 0.91       | 64.03 ± 0.56       |
> | General-purpose multimodal | MT            | 81.29 ± 0.92     | 39.20 ± 2.96      | 62.31 ± 1.24       | 66.30 ± 1.12       | 69.24 ± 0.34       |
> | multi-graph        | ***AMIGO***        | 79.23 ± 1.04     | 38.92 ± 1.32      | 58.68 ± 2.12       | 65.73 ± 0.89       | 70.42 ± 0.79       |
> | multi-graph        | ***MaxCorrMGNN***  | 77.43 ± 1.35     | 35.97 ± 2.73      | 56.23 ± 1.32       | 63.72 ± 1.64       | 71.46 ± 1.01       |
> | multi-graph        | ***MGLAM***        | 81.29 ± 0.96     | 38.93 ± 1.02      | 61.96 ± 1.04       | 62.29 ± 1.29       | 69.52 ± 0.39       |
> | Proposed model             | **MGMT**      | **83.11 ± 0.84** | **42.13 ± 2.52**  | **65.47 ± 2.39**   | **69.90 ± 1.19**   | **73.21 ± 0.59**   |

---

> > ### Author Response · Authors · 2025-11-24
> > **Rebuttal, Continued**
> >
> > These additions ensure that MGMT is directly compared to SOTA graph and multi-graph architectures, not only to generic multimodal fusion models. Across all settings, MGMT outperforms these graph-based baselines, supporting the claim that its meta-graph–based, subgraph-level fusion provides benefits beyond what existing GT and multi-graph models achieve on these neuroscience and Alzheimer tasks. We retain the multimodal baselines (e.g., MedFuse, FlexCare) to show that even when competing against strong vector-level fusion schemes commonly used in clinical applications, graph-native MGMT remains competitive without changing the underlying datasets.
> >
> > >**W3, Q2:** The authors claim that the graph-specific Transformer encoders produce depth-aware intra-graph representations. How does the depth-aware GT layers differ from existing multi-layer aggregation techniques?
> >
> > We appreciate the reviewer’s thoughtful comment. While the depth-aware approach is discussed in detail in Appendix A2, it is only one component of our backbone-agnostic framework and fits naturally into MGMT’s meta-graph pipeline. Theoretically, Section 4.1 shows that this aggregation realizes an $L$-hop mixing operator for Graph Transformers, enlarging the class of functions that can be approximated without increasing depth and explaining its consistent performance gains in practice. We have also added new experiments (Fig. 6), which provide a layer-wise visualization of attention and depth-confidence scores on the LFP data: high-confidence layers concentrate edge attention and node-summed weights on distal CA1 electrodes (the behaviorally informative subnetwork identified in Fig. 5), whereas low-confidence layers exhibit more diffuse, non-specific patterns. Together, these results indicate that this component of MGMT is not merely averaging multi-layer embeddings, but is learning task-relevant, depth-specific structural semantics that *provide additional guidance* to the meta-graph module, directly impacting which layers, and thus which supernodes and meta-graph connections, are emphasized in MGMT’s final predictions.
> >
> >
> >
> > >**Q2:** In section 4.2, What do you mean multimodal classification? Also, in Figure 3, what’s the meaning of Modality 1, 2,…?
> >
> > To avoid ambiguity regarding our simulation process, we have revised the sentence to: “We create an intermediate binary label for each modality, then aggregate them into an entity-level label by applying a threshold to a weighted sum of these modality-specific labels.” This explicitly clarifies that the prediction task is to classify each entity (sample). We also refined the description around Figure 3 to note that each “Modality $m$” corresponds to one of the five simulated graphs associated with the same entity, each generated with its own feature and noise configuration.

---

### Official Review · Reviewer_PY7g · 2025-10-30

**Soundness:** 3
**Presentation:** 3
**Contribution:** 3
**Rating:** 6
**Confidence:** 3

**Summary:**

This paper proposes the Multi-Graph Meta-Transformer (MGMT), a unified, scalable, and interpretable framework for cross-graph learning, addressing the key challenge of effectively integrating information across heterogeneous graphs (with differing topologies, scales, semantics, and often no shared node identities). MGMT’s core pipeline includes three stages: (1) applying Graph Transformer (GT) encoders to each graph, with depth-aware aggregation of layer outputs to generate intra-graph latent representations; (2) extracting task-relevant supernodes via attention scores and constructing a meta-graph that connects functionally aligned supernodes across graphs using latent similarity; (3) applying additional GT layers on the meta-graph to enable joint reasoning over intra- and inter-graph structures. Evaluated on three synthetic datasets and two real-world neuroscience applications (hippocampal LFP-based stimulus prediction and Alzheimer’s disease detection), MGMT consistently outperforms state-of-the-art baselines (e.g., Meta-Transformer, MMGL) in graph-level prediction tasks while providing interpretable representations (via supernodes and superedges) that facilitate scientific discoveries.

**Strengths:**

This paper tackles a critical gap in multi-graph learning: existing fusion methods either assume node alignment (limiting applicability to heterogeneous graphs) or collapse graph topology into a single vector (losing structural information), while MGMT preserves both intra- and inter-graph structure—a highly valuable contribution.

The proposed framework is notably versatile, unifying three common multi-graph scenarios (multi-modal, multi-view, multi-subject) under one umbrella, which broadens its practical utility across domains like neuroscience, molecular biology, and social networks.

MGMT is supported by solid theoretical foundations: the paper formally proves that MGMT’s depth-aware GTs enable L-hop mixing (surpassing vanilla GTs) and that MGMT achieves a smaller approximation error than late fusion methods, enhancing the credibility of its design.

Built-in interpretability is a key strength: supernodes highlight influential substructures (e.g., distal CA1 electrodes in neuroscience) and superedges reveal cross-graph alignments, making MGMT not only a predictive tool but also a means to drive scientific insights—rare in many graph learning models.

The experimental design is thorough: it covers synthetic datasets (varying noise, node counts, and sample sizes) and real-world neuroscience tasks, includes comprehensive ablations (validating adaptive depth, supernode selection, and inter-graph edges), and compares against diverse baselines (single-source, early fusion, and state-of-the-art multimodal models), leaving little doubt about MGMT’s effectiveness.

**Weaknesses:**

MGMT relies on fixed thresholds (τ for supernode extraction, γ for inter-graph edge construction) that require cross-validation to tune. This manual thresholding lacks adaptability to dynamic or unseen data distributions, and the paper does not explore alternatives like learnable edge weights or data-driven dynamic thresholds, which could improve robustness.

While the paper acknowledges rising computational complexity with more modalities/larger graphs (due to attention layers’ quadratic cost), it only mentions sparse/low-rank attention as a future direction without providing preliminary optimization results (e.g., runtime comparisons with sparse attention variants). This limits confidence in MGMT’s scalability for large-scale multi-graph scenarios (e.g., hundreds of graphs with thousands of nodes).

The attention-based importance scores for supernode selection may fail to capture causal structures in noisy data—a limitation noted in the paper—but no mitigating strategies (e.g., robust attention, causal inference integration) are discussed or explored, leaving a gap in addressing real-world data imperfections.
Experimental validation is concentrated on neuroscience applications. Although the paper claims MGMT is applicable to molecular biology and social networks, it provides no empirical results in these domains, weakening the evidence for its cross-domain generality.

**Questions:**

1. Why did you choose fixed thresholds (τ and γ) for supernode extraction and inter-graph edge construction instead of more adaptive approaches (e.g., learnable edge weights, dynamic thresholding based on data statistics)? How does threshold tuning affect MGMT’s performance across datasets with varying graph heterogeneity?
2. For the computational complexity issue (quadratic in graph size/node count), have you conducted preliminary experiments with optimization techniques like sparse attention or low-rank attention? If so, could you share runtime and accuracy trade-off results; if not, what is the feasibility of these techniques for MGMT?
3. Given that attention-based supernode scores struggle with causal structure in noisy data, do you have plans to integrate causal inference methods (e.g., causal attention, intervention-based importance scoring) to improve supernode selection accuracy? Could you outline potential design modifications for this?
4. The paper asserts MGMT’s applicability to molecular biology and social networks, but only validates it in neuroscience. Can you provide preliminary experimental results in these other domains (e.g., molecular property prediction, social network classification) or explain how neuroscience findings generalize to them, given differences in graph structure (e.g., molecular graphs vs. brain connectivity graphs)?

---

> ### Author Response · Authors · 2025-11-20
>
> We thank the reviewer for the constructive feedback on our work. We hope the proposed revisions and clarifications would address the concerns raised.
> >Why did you choose fixed thresholds..
>
> We apologize for the lack of clarity and have addressed this issue in the revised version. The model does learn edge weights, which are subsequently used in the thresholding step. As discussed in Sec. 3.1.2 and Appendix A10, MGMT’s performance remains stable over a reasonably wide range of $\tau$ and $\gamma$, with degradation only at extreme values. These thresholds are not fixed; rather, they are treated as regular hyperparameters and tuned on validation data. However, we agree that a dynamic thresholding strategy could further improve flexibility. We have therefore added a new experiment in Appendix A13 in which $\tau$ and $\gamma$ are replaced by data-driven, distribution-based thresholds derived from the empirical score distributions. More specifically, they are set dynamically using quantiles of the observed attention-score distributions. Empirically, the dynamic variant of MGMT achieves results comparable to the validation-tuned version, confirming that the model does not depend on finely tuned absolute thresholds and that its performance is robust under fully data-driven sparsification rules.
> >For the computational complexity issue ...
>
> We agree that using fully-connected Graph Transformer backbones leads to quadratic complexity in the number of nodes and can become prohibitive for very large graphs. Our original complexity discussion implicitly assumed dense attention; in the revised version, we have clarified this in Appendix A9.1 by explicitly distinguishing different backbone regimes. In particular, the main MGMT implementation in Section~3.1.1 uses a GAT-style localized attention backbone, where each node only attends to its neighbors. For a graph $G_i$ with $|E_i|$ edges and $d$-dimensional features, this yields a per-layer cost of $O(|E_i| d)$, and a total cost of $O(\sum_{i=1}^n |E_i| d)$ across $n$ graphs, which is effectively near-linear in the graph size for the sparse graphs considered in our experiments.
> By contrast, dense attention backbones (e.g., fully-connected Graph Transformers) incur $O(N_i^2 d)$ cost per graph with $N_i$ nodes, while sparse attention variants (e.g., top-$K$ or expander-based attention) reduce this to $O(N_i K d)$ with $K \ll N_i$. MGMT is backbone-agnostic and can directly benefit from such sparse or low-rank attention mechanisms: they only change the cost of the per-graph encoder, while the meta-graph construction remains unchanged and scales with the number of selected supernodes. The revised complexity analysis (A9.1) and backbone comparisons (A12) indicate that MGMT can leverage existing sparse/linear GT designs to mitigate quadratic costs in regimes where dense attention would be infeasible. To make this concrete, we added a new experiment in Appendix A15 where we replace dense attention in the graph-specific encoder with a simple top-$k$ sparse variant and report the resulting runtime–accuracy trade-offs.
> >Given that attention-based supernode scores struggle with causal structure in noisy data..
>
> We note that our original discussion on limitations was overly concise and may have been confusing. To clarify: attention-based importance scores inherently capture correlations rather than causal relationships, regardless of noise levels. However, noisy settings exacerbate this limitation by introducing spurious correlations that can dominate attention weights, making it even more challenging to identify causally important nodes.
> Recent advances in causal graph learning offer promising directions for identifying causally important nodes. Causal masking approaches such as CAL (Sui et al., 2022) learn to disentangle causal from spurious features through intervention-based training. Counterfactual methods reviewed in Guo et al. (2025) identify critical graph components by measuring prediction changes under minimal graph edits. In Appendix A14, we present one concrete potential design using CAL, while future work can explore additional directions for causal learning in multi-graph settings.
> >The paper asserts MGMT’s applicability to molecular biology..
>
> We appreciate the reviewer’s comment regarding the broader applicability of our method. While our experiments focus on neuroscience to demonstrate the practical utility of MGMT, the framework itself is not domain-specific and can be applied to any scientific setting involving graph-structured data. The two neuroscience datasets we study are structurally and semantically distinct, illustrating MGMT’s flexibility across heterogeneous graph types (electrophysiological vs. clinical/imaging data). These differences highlight MGMT’s capacity to generalize across domains with diverse graph structures. In parallel, we are exploring applications to mobile health data, and extending MGMT to such settings is part of our future work.

---

### Official Review · Reviewer_UCot · 2025-11-11

**Soundness:** 3
**Presentation:** 3
**Contribution:** 2
**Rating:** 4
**Confidence:** 2

**Summary:**

This paper proposes a unified framework for multi-graph learning, termed Multi-Graph Meta-Transformer (MGMT). Its core innovation lies in encoding multiple heterogeneous graphs using Graph Transformers, extracting representative supernodes via an attention mechanism, and constructing a meta-graph that captures inter-graph dependencies, thereby enabling information exchange and alignment among graphs within a shared latent space.

**Strengths:**

By stacking additional Transformer layers on the meta-graph, the model learns both intra- and inter-graph structural features, achieving depth-aware aggregation and hierarchical reasoning. This design improves the precision and robustness of cross-graph representations and provides intrinsic interpretability, leading to superior performance over existing fusion models in neuroscience and multimodal medical data analysis tasks.

**Weaknesses:**

The title only reflects the method name “Multi-Graph Meta-Transformer” and does not convey the core innovation or research objective, such as “cross-graph fusion” or “interpretable multimodal graph learning.” This makes the title insufficient to highlight the study’s contributions. It is recommended that the authors revise the title to better communicate the paper’s novelty and focus.


The positioning of the innovation is somewhat ambiguous. The differences between MGMT and existing Meta-Transformer and multimodal fusion models are insufficient to clearly define it as a “new framework.” MGMT is highly similar in naming and overall concept to Meta-Transformer (Ma et al., 2022). Although the paper introduces the concepts of meta-graph and supernodes, the approach essentially remains a variant of “cross-modal Transformer + graph structure reconstruction.” The paper does not adequately demonstrate substantial theoretical or representational improvements over previous work.


Minor language and formatting issues.
Section 2.1.1: “matrices and and bold lowercase …” – redundant “and.”
Proof of Theorem 3.4: “the desired results follows …” – subject-verb disagreement; should be results follow.
Appendix A1: “Shi et al. Shi et al. (2020)” – duplicated citation; should be Shi et al. (2020).
The cited literature is relatively outdated, lacking references to recent studies (within the past two years) on multi-graph learning and cross-modal Transformers. The experimental baselines are also old, mainly consisting of earlier methods and self-ablation experiments, which fail to demonstrate the method’s advancement and competitiveness.


Section A6 mainly describes the setup and results of the ablation experiments but does not analyze in detail how the absence of each module leads to performance degradation. It lacks an interpretation of the experimental results and explanations from a mechanistic perspective. It is suggested to include failure analysis and visualization to strengthen the discussion.


The paper does not provide a clear explanation for the runtime fluctuation observed in Figure A8 as the attention threshold τ increases, especially in the left panel where τ controls supernode selection. This non-monotonic trend is inconsistent with the theoretically smooth growth of MGMT’s computational complexity described in the paper. The authors are advised to further discuss possible reasons for this phenomenon.

**Questions:**

The experiments only verify graph-level classification tasks and lack extended experiments for node-level or edge-level tasks, which limits the demonstration of MGMT’s applicability to other graph learning scenarios such as link prediction, anomaly detection, or graph generation.

---

> ### Author Response · Authors · 2025-11-20
> **Rebuttal**
>
> We thank the reviewer for the constructive feedback on our work. We hope the proposed revisions and clarifications would address the concerns raised.
> >**W1:** The title only reflects the method name “Multi-Graph Meta-Transformer” and does not convey the core innovation or research objective...
>
> We thank the reviewer for this helpful suggestion.  In the revised version, we plan to update the title to better highlight both the cross-graph fusion aspect and the interpretability provided by our attention-selected meta-graph construction. As per ICLR policy, title changes are allowed in the post-rebuttal revision stage.
> > **W2:** The positioning of the innovation is somewhat ambiguous..
>
> That is a great point. In the revision, we have made this distinction explicit in the Introduction and in the new multi-graph learning subsection of the Related Work. We have also added new experiments comparing MGMT against additional SOTA multi-graph and transformer models to provide empirical evidence of its advantage over existing methods. As clarified in the revision, although MGMT’s high-level objective may resemble other methods, it is explicitly designed for the multi-graph setting, where each entity is represented by heterogeneous graphs with unaligned node sets. Its core innovation is performing fusion at the subgraph level via a meta-graph of attention-selected supernodes, rather than through pooled graph embeddings or shared context tokens. As a result, MGMT not only improves predictive performance in the heterogeneous multi-graph setting, but also enables tracing which cross-graph interactions the model uses and how they contribute to the task. These capabilities are not offered by prior fusion schemes that collapse each graph into a single vector or treat modalities as unordered tokens.
> > **W3:** Minor language and formatting issues.
>
> We thank the reviewer for carefully pointing out these formatting issues and typos. In the revised manuscript, we have corrected the mentioned typos.

---

> ### Author Response · Authors · 2025-11-24
> **Rebuttal, Continued 1**
>
> > **W4:** The cited literature is relatively outdated... The experimental baselines are also old, mainly consisting of earlier methods and self-ablation experiments...
>
> We appreciate this comment and have substantially updated both the related work and experimental baselines to better reflect recent advances. In the revised Related Work, we now explicitly covered: (1) modern graph transformer backbones (GraphGPS, GRIT, Exphormer, EGT, and HDSE), as state-of-the-art single-graph GTs, to make clear that MGMT is backbone-agnostic and to motivate the additional backbone experiments in Appendix A12; (2) entity-level multi-graph models (AMIGO, MaxCorrMGNN, and MGLAM), which take multiple graphs per entity and are therefore the most comparable to our setting; these are now included as strong baselines in Results section; (3) multi-view GTs with aligned nodes (MGT and MVGT/MVGTrans), cited to clarify how their aligned multi-view assumption differs from MGMT’s heterogeneous, unaligned multi-graph regime; and (4) population-level multi-graph and graph-of-graphs approaches (AMGL, GraphFM, GraphAlign, and SamGoG), used to position MGMT as a complementary, entity-level meta-graph framework. On the experimental side, we have substantially expanded the evaluation to include both recent multi-graph baselines and modern GT backbones. First, we incorporate AMIGO, MaxCorrMGNN, and MGLAM as strong multi-graph baselines, with results summarized in Table A2 (and visualized in Figs. 3–4) and referenced in the main Results section, also shown below:
>
> **Table A2.** Accuracy (mean ± standard error) for new SOTA GT-based multi-graph baselines across datasets.
> The proposed model is shown in **bold**.
>
> | Category                      | Model         | Alzheimer        | LFP Data          | Experiment 1       | Experiment 2       | Experiment 3       |
> |------------------------------|---------------|------------------|-------------------|--------------------|--------------------|--------------------|
> | multi-graph       | AMIGO         | 79.23 ± 1.04     | 38.92 ± 1.32      | 58.68 ± 2.12       | 65.73 ± 0.89       | 70.42 ± 0.79       |
> | multi-graph       | MaxCorrMGNN   | 77.43 ± 1.35     | 35.97 ± 2.73      | 56.23 ± 1.32       | 63.72 ± 1.64       | 71.46 ± 1.01       |
> | multi-graph       | MGLAM         | 81.29 ± 0.96     | 38.93 ± 1.02      | 61.96 ± 1.04       | 62.29 ± 1.29       | 69.52 ± 0.39       |
> | Proposed model               | **MGMT**      | **83.11 ± 0.84** | **42.13 ± 2.52**  | **65.47 ± 2.39**   | **69.90 ± 1.19**   | **73.21 ± 0.59**   |
>
> These additions ensure that MGMT is directly compared to SOTA multi-graph architectures, not only to generic multimodal fusion models. Across all settings, MGMT outperforms these graph-based baselines, supporting the claim that its meta-graph–based, subgraph-level fusion provides benefits beyond what existing multi-graph models achieve. We retain the multimodal baselines (e.g., MedFuse, FlexCare) to show that even when competing against strong vector-level fusion schemes commonly used in clinical applications, graph-native MGMT remains competitive without changing the underlying datasets.

---

> > ### Author Response · Authors · 2025-11-24
> > **Rebuttal, Continued 2**
> >
> > Additionally, in Section A12, we conducted a systematic backbone study in which we replaced the original GAT encoder with several state-of-the-art Graph Transformer variants (GraphGPS, GRIT, Exphormer, EGT), evaluating them both as single-graph models (Table A6) and as backbones inside MGMT (Tables A7 and shown below) to assess MGMT’s sensitivity to backbone choice, disentangle the gains from stronger GT layers versus depth-aware aggregation, and demonstrate that MGMT maintains robust performance  (Tables A7 and shown below) and runtime (Table A8) across diverse GT backbones. These experiments show that while stronger GT backbones bring modest improvements in the single-graph setting, depth-aware aggregation closes most of this gap, and in the multi-graph regime, MGMT with depth-aware GAT, GraphGPS, GRIT, and Exphormer attains very similar accuracy and runtime.
> >
> > **Table A7.** Test accuracy (mean ± standard error) for MGMT variants using different depth-aware GT backbones for feature encoding and final prediction. The default MGMT (depth-aware GAT) row is in **bold**; per-column best values are also **bolded**.
> >
> > | Model                             | LFP Dataset       | Alzheimer        | Experiment 1     | Experiment 2     | Experiment 3     |
> > | --------------------------------- | ----------------- | ---------------- | ---------------- | ---------------- | ---------------- |
> > | GAT                               | 40.64 ± 0.223     | 81.20 ± 0.85     | 64.20 ± 2.40     | 68.80 ± 1.17     | 71.45 ± 0.57     |
> > | **Depth-aware GAT aggregation**   | **42.13 ± 0.252** | 83.11 ± 0.84     | **65.47 ± 2.39** | **69.90 ± 1.19** | **73.21 ± 0.59** |
> > | Depth-aware GraphGPS aggregation  | 41.94 ± 0.162     | 80.12 ± 0.67     | 62.37 ± 1.56     | 67.49 ± 0.98     | 72.38 ± 0.83     |
> > | Depth-aware GRIT aggregation      | 42.04 ± 0.345 | 82.59 ± 1.03     | 61.06 ± 2.31     | 69.23 ± 1.01     | 72.98 ± 0.61     |
> > | Depth-aware Exphormer aggregation | 41.86 ± 0.427     | **83.29 ± 0.58** | 62.93 ± 1.92     | 67.46 ± 1.32     | 72.46 ± 0.49     |
> > | Depth-aware EGT aggregation       | 40.08 ± 0.162     | 81.79 ± 1.26     | 60.52 ± 1.86     | 68.96 ± 0.94     | 71.30 ± 0.49     |
> >
> > Together, these results demonstrate that MGMT’s gains stem primarily from its supernode/meta-graph fusion rather than any particular GT layer, and that the framework is effectively **backbone-agnostic** and can be straightforwardly instantiated with new GT backbones as they are introduced in the future.
> > >**W5:**  Section A6 mainly describes the setup and results of the ablation experiments...
> >
> > We thank the reviewer for this suggestion. In the revised version, Appendix A7 (ABLATION STUDY) has been expanded to go beyond ablation results: for each component (adaptive depth, supernode selection, inter-/intra-graph edges, and the meta-graph), we provide a mechanistic interpretation of how its removal alters the meta-graph and degrades performance (Table A3, Figure 2). Specifically, depth ablation is linked to learned layer confidences and attention patterns (Figure 6), and edge ablations to similarity-threshold sensitivity and smoothness analyses (Appendices A5 and A10). These additions offer a clearer failure analysis and demonstrate the necessity of each module for MGMT’s performance.
> > >**W6:**  The paper does not provide a clear explanation for the runtime fluctuation observed in Figure A8...
> >
> > The runtime and accuracy curves in Figure A8 were computed over 10 repetitions, introducing variance due to the stochastic nature of attention-based node selection and random initialization. To address this, we repeated the sensitivity analysis using 50 independent runs per threshold value. The updated results (Appendix A10) now show a much smoother, monotonic decline in runtime with increasing $\tau$, consistent with expectations.
> > >**Q1:** The experiments only verify graph-level classification tasks...
> >
> > We appreciate the reviewer’s thoughtful comment. In this work, we intentionally focus on graph-level prediction tasks, which align with our primary motivating applications, including neural decoding and Alzheimer’s prediction. As clarified in the revised manuscript, MGMT is specifically designed for settings where each entity is associated with a label and a set of heterogeneous graphs. That said, we agree that extending MGMT to support tasks such as node classification and link prediction represents an interesting direction for future work.

---

### Author Response · Authors · 2025-11-20

We thank all reviewers for their time and thoughtful evaluation of our work, as well as for their constructive feedback. We have uploaded a revised version of the manuscript in which all updated sections are highlighted in blue.

---

### Author Response · Authors · 2025-12-03
**Summary of Discussion and Changes**

We would like to thank the reviewers for the time and care they devoted to evaluating our work, as well as for providing thoughtful, constructive, and encouraging feedback. In response, we have revised the manuscript by (1) clarifying the motivation and contributions, (2) strengthening the positioning within multi-graph learning and Graph Transformer literature, and (3) adding additional experiments, analyses, and theoretical justifications to better support the main claims. For ease of assessment, we summarize the key changes below.

1. **Clarified the motivation, setting, and contributions in the Introduction.**
   We now state explicitly the core limitation of prior fusion approaches in the heterogeneous multi-graph regime (unaligned node sets), namely the lack of a structure-preserving and interpretable mechanism that captures both *intra-graph* topology and *inter-graph* cross-structure interactions; we also added a clearer, concise contributions summary.

2. **Rewrote and reorganized Related Work into a dedicated standalone section.**
   We added a new multi-graph learning subsection to the Literature Review section, covering entity-level multi-graph models, multi-view GTs with aligned nodes, and population-level graph-of-graphs approaches. In addition, we substantially expanded discussion of modern Graph Transformer literature to include global, sparse, structure-aware, and hierarchical GT variants.

3. **Strengthened the “depth-aware” component with additional theory and evidence.**
   We expanded the depth-aware aggregation description (Appendix A2), clarified how it differs from purely heuristic layer averaging, and added layer-wise analyses and visualizations of learned depth confidence and attention patterns (e.g., Fig. 6). Furthermore, we added a formal theoretical discussion relating $L$-hop mixing to WL expressivity (Appendix A4.3).

4. **Added strong multi-graph baselines and direct comparisons.**
   In response to the reviewers’ comments, we extended our already extensive set of baseline models by adding several additional entity-level multi-graph methods (e.g., AMIGO, MaxCorrMGNN, MGLAM) across all datasets and summarizing their results alongside MGMT (Table A2, Fig. 3, and Fig. 4). The updated manuscript now evaluates MGMT against a comprehensive collection of state-of-the-art multi-graph architectures.

5. **Added a systematic GT-backbone study and runtime breakdowns.**
   While the original submission primarily implemented MGMT with a GAT-style (localized-attention) encoder, the revised version substantially broadens this by incorporating multiple modern Graph Transformer (GT) backbones. Specifically, we evaluated GraphGPS, GRIT, Exphormer, and EGT both as single-graph baselines (Table A6) and as drop-in backbones within MGMT (Table A7), allowing us to test backbone sensitivity and disentangle improvements due to stronger encoders from those attributable to MGMT’s meta-graph fusion itself (Appendix A12). We additionally provide a per-epoch runtime breakdown across MGMT variants (Table A8) to quantify computational trade-offs and to demonstrate that MGMT’s performance gains are not tied to the original GAT choice but persist even under stronger GT backbones.

6. **Improved robustness and sensitivity reporting for thresholding and sparsification.**
   We have extended and improved our sensitivity analysis (Appendix A10; Fig. A8), showing that MGMT does not rely on finely tuned absolute thresholds by introducing a new *dynamic, distribution-based thresholding* variant that sets $\tau$ and $\gamma$ automatically using quantiles of the empirical attention/similarity score distributions (Appendix A13). This fully data-driven sparsification achieves performance comparable to the validation-tuned setting, supporting the conclusion that MGMT is robust to threshold choices and does not depend on precise manual calibration of $\tau$ and $\gamma$.

7. **Expanded ablations into a clearer failure analysis.**
   The ablation section has been expanded to provide mechanistic interpretations of how removing each component (adaptive depth, supernode selection, intra- and inter-graph edges, meta-graph) alters the learned meta-graph structure and degrades performance (Appendix A7 and related Fig. 2).

We believe these revisions address the reviewers' concerns and substantially improve the manuscript in terms of clarity, organization, and both the theoretical and empirical support for our claims. We thank the Area Chair for their time and consideration.

---

### Meta-Review · Area_Chair_J3nH · 2025-12-25

**Summary:**

The reviewers have raised the following major concerns:

(1) Novelty: The differences between MGMT and existing Meta-Transformer and multimodal fusion models are insufficient to clearly define it as a new framework.

(2) Motivation in relation to the drawbacks of existing approaches.

(3) Complexity and scalability.

(4) Important baselines are missing from the experiments.

**Reviewer Concerns:**

In the revised manuscript, the authors have added new numerical results, included a discussion of computational complexity, and provided higher-level conceptual explanations addressing several of the reviewers’ comments. These additions help clarify certain aspects of the method and partially address some previously raised concerns.

Nevertheless, after carefully reading the revised paper and the accompanying rebuttal, I share Reviewer UCot’s view that the issue of novelty remains a central concern and has not been fully resolved. The proposed approach appears to primarily combine well-established graph transformer architectures with multi-modal fusion techniques, and it is not entirely clear how this combination constitutes a fundamentally new framework rather than an integration or extension of existing methods.

In particular, based on Section 3 and the theoretical analysis in Appendices A3 and A4, the reported theoretical performance gains seem to stem mainly from more elaborate extensions of known mechanisms. For example, in the proof of Theorem 4.3 (Appendix A3), the choice of $f(\cdot)$ as the identity function suggests that the model reduces to a more complex variant of GAT with additional learnable parameters, which aligns with the observations made by Reviewer dTh6. A similar impression arises in the comparison with Zhang et al. (2023). Therefore, the increased model complexity, especially in the worst-case scenario, is an expected consequence as a trade-off.

Moreover, the graph fusion component relies on heuristic construction based on feature cosine similarity. Potential errors introduced during feature generation or heuristic graph construction may propagate through subsequent components of the model, yet this issue is not analyzed in sufficient depth.

Overall, while the revisions improve the presentation and address some concerns, I remain unconvinced that the level of methodological novelty and insight meets the bar expected for ICLR.

**Reviewer Scores:**

I am not aware of any active discussion in the rebuttal phase, and it is hard to assess whether the reviewers will revise their scores. However, as explained above, there are key issues that remain unresolved.

---

### Decision · Program_Chairs · 2026-01-26

Reject